# A semiconducting layered metal-organic framework magnet

Chongqing Yang[1,2], Renhao Dong [1], Mao Wang [3], Petko St. Petkov [4], Zhitao Zhang [3], Mingchao Wang[1], Peng Han [5], Marco Ballabio[5], Sascha A. Bräuninger[6], Zhongquan Liao[7], Jichao Zhang [8], Friedrich Schwotzer[1], Ehrenfried Zschech[7], Hans-Henning Klauss[6], Enrique Cánovas[5,9], Stefan Kaskel [1], Mischa Bonn[5], Shengqiang Zhou [3], Thomas Heine [1,3,10] & Xinliang Feng[1,2]

The realization of ferromagnetism in semiconductors is an attractive avenue for the development of spintronic applications. Here, we report a semiconducting layered metal-organic framework (MOF), namely $K_3Fe_2[(2,3,9,10,16,17,23,24$-octahydroxy phthalocyaninato)Fe] ($K_3Fe_2[PcFe-O_8]$) with spontaneous magnetization. This layered MOF features in-plane full $\pi$-$d$ conjugation and exhibits semiconducting behavior with a room temperature carrier mobility of $15 \pm 2 \, cm^2 \, V^{-1} \, s^{-1}$ as determined by time-resolved Terahertz spectroscopy. Magnetization experiments and $^{57}Fe$ Mössbauer spectroscopy demonstrate the presence of long-range magnetic correlations in $K_3Fe_2[PcFe-O_8]$ arising from the magnetic coupling between iron centers via delocalized $\pi$ electrons. The sample exhibits superparamagnetic features due to a distribution of crystal size and possesses magnetic hysteresis up to 350 K. Our work sets the stage for the development of spintronic materials exploiting magnetic MOF semiconductors.

[1] Center for Advancing Electronics Dresden (cfaed) & Department of Chemistry and Food Chemistry, Technische Universität Dresden, Mommsenstrasse 4, 01062 Dresden, Germany. [2] School of Chemistry and Chemical Engineering, Shanghai Jiao Tong University, Dongchuan Road 800, 200240 Shanghai, China. [3] Helmholtz-Zentrum Dresden-Rossendorf, Institute of Ion Beam Physics and Materials Research, Bautzner Landstr. 400, 01328 Dresden, Germany. [4] University of Sofia, Faculty of Chemistry and Pharmacy, J. Bourchier blvd. 1, 1164 Sofia, Bulgaria. [5] Max Planck Institute for Polymer Research, Ackermannweg 10, 55128 Mainz, Germany. [6] Institut für Festkörper und Materialphysik, Technische Universität Dresden, 01062 Dresden, Germany. [7] Fraunhofer Institute for Ceramic Technologies and Systems (IKTS), 01109 Dresden, Germany. [8] Shanghai Synchrotron Radiation Facility, Shanghai Advanced Research Institute, Chinese Academy of Sciences, No.239 Zhangheng Road, Shanghai 201204, China. [9] Instituto Madrileño de Estudios Avanzados en Nanociencia (IMDEA Nanociencia), Faraday 9, 28049 Madrid, Spain. [10] Wilhelm-Ostwald-Institute of Physical and Theoretical Chemistry, Leipzig University, Linnéstr. 2, 04103 Leipzig, Germany. Correspondence and requests for materials should be addressed to R.D. (email: renhao.dong@tu-dresden.de) or to X.F. (email: xinliang.feng@tu-dresden.de)

Since the 1980s, spintronics has been intensely researched for the next generation of data operation and storage in information technology, because spintronics could offer an exciting prospect of combining the semiconductor-based information operation (logic) and magnetic based data storage (memory) into the same device[1–4]. However, the practical realization of spintronic devices remains challenging due to the lack of functional materials displaying both, semiconducting and high-temperature magnetic ordering properties[5–7]. Historically, the research of ferromagnetic semiconductor materials has been primarily focused on the development and characterization of inorganic dilute magnetic semiconductors (DMSs), such as $p$-type Mn-doped II–VI[8], III–V[9], and IV–VI semiconductors[10,11]. Recently, intrinsic ferromagnetism has been demonstrated in layered inorganic two-dimensional (2D) materials, such as chromium germanium telluride ($Cr_2Ge_2Te_6$)[12] and chromium triiodide ($CrI_3$)[13]. Despite enabling long-range electronic coupling via short metal-metal bonds in these inorganic magnets[14], the low magnetic ordering temperature (e.g., $Tc = 159$ K for GaMnAs, 61 K for $Cr_2Ge_2Te_6$, and 68 K for $CrI_3$, see Supplementary Table 1), together with the limited chemical tunability present major drawbacks for their practical applications.

Metal-organic frameworks (MOFs) based on crystalline coordination polymers have been regarded as promising organic magnets[15]. These materials hold promise because the magnetic behavior can be tailored at the molecular scale through tuning their constituent organic ligands, metal centers (spin carriers) and even filling functional guest molecules in porous MOFs[16]. Magnetic ordering at low temperature ($Tc < 180$ K, seen in Supplementary Table 2) has been already proven in bulk (3D) MOFs[14]. However, most of them behave as electrical insulators ($\sigma < 10^{-12}$ s m$^{-1}$) because of the localized molecular orbitals and minimal band dispersion in their intrinsic electronic structures, thus hampering their potential applications in spintronics/electronics. Taking advantage of the porous structure, hosting guest molecules (like tetracyanoquinododimethane) in MOFs has recently been shown to significantly enhance the electrical conductivity[17], due to the charge transfer of guests to the metal nodes or organic linkers. In addition, linking redox-active ligands, such as 2,5-dihydroxybenzoquinone[14,18], pyrazine[19], could also generate long-range charge transport and strong magnetic exchange, leading to conductivities as high as 20 s m$^{-1}$ and magnetic ordering. On the other hand, since the first report in 2012[20], conjugated MOFs with layered structures have been developed displaying high electrical conductivities (up to $10^5$ s m$^{-1}$). This class of materials has typically been constructed by linking N, O, or S ortho-disubstituted benzene, triphenylene or coronene ligands with transition metal ions, and featured with full $\pi$-$d$ conjugation in 2D planes[20–27]. Furthermore, density functional calculation (DFT) has predicted that strong planar $\pi$-$d$ hybridization combined with large magnetic anisotropy could generate magnetic ordering at room temperature[28]. However, the simultaneous realization of room temperature spontaneous magnetization and semiconducting behavior in a MOF has not been experimentally demonstrated to date.

Here, we report a conjugated $K_3Fe_2[PcFe–O_8]$ MOF with square lattice geometry and interplane van der Waals (vdW) layer-stacking structure by employing (2,3,9,10,16,17,23,24-octahydroxy phthalocyaninato) iron (PcFe-OH$_8$) as ligand and square planar iron-bis(dihydroxy) complex as linkage as well as K$^+$ as the counter ions. The resultant $K_3Fe_2[PcFe–O_8]$ exhibits a typical semiconducting behavior with a room temperature carrier mobility of $15 \pm 2$ cm$^2$ V$^{-1}$ s$^{-1}$ as determined by time-resolved Terahertz spectroscopy (TRTS). Moreover, this $K_3Fe_2[PcFe–O_8]$ presents magnetic correlation in nanoscale crystallites, which overall shows superparamagnetism with a broad distribution of blocking temperatures. DFT calculations reveal that the strong hybridization of the $\pi$-$d$ orbitals and the high concentration of Fe spins (0.7 $\mu_B$ per Fe site) in $K_3Fe_2[PcFe–O_8]$ contribute to its high-temperature magnetic exchange interactions. By further optimizing the crystalline quality and increasing the crystallite size, our work presents the possibility to achieve room temperature ferromagnetism in a semiconducting layered MOF, highlighting the potential for developing a new generation of MOFs-based spintronics.

## Results

**Synthesis and structural analysis of $K_3Fe_2[PcFe–O_8]$.** The $K_3Fe_2[PcFe–O_8]$ MOF with potassium cation (K$^+$) as counter ions was synthesized through the coordination reaction between PcFe-OH$_8$ and iron (II) acetate (Fe(OAc)$_2$) in the presence of potassium acetate (KOAc) (Fig. 1a, Supplementary Fig. 1, see Supplementary Methods). Fourier transform infrared spectra (FTIR, Supplementary Fig. 2) showed that O–H stretching bonds (~3300 cm$^{-1}$) disappeared in $K_3Fe_2[PcFe–O_8]$, indicating the successful coordination reaction between the –OH groups and Fe ions. Besides, new peaks centered at 534 and 468 cm$^{-1}$ can be assigned to O–Fe–O stretching bonds. Energy-dispersive X-ray (EDX, Supplementary Figs. 3–4) spectroscopy and X-ray photoelectron spectroscopy (XPS, Supplementary Fig. 5) were carried out to analyze the composition of $K_3Fe_2[PcFe–O_8]$, which revealed the presence of C, N, O, Fe and K in the MOF sample. The core level spectrum of Fe ($2p$) contains peaks at 710.8, 713.4, 724.2, 726.6 eV, attributable to the Fe$_{(II)}$ $2p_{3/2}$, Fe$_{(III)}$ $2p_{3/2}$, Fe$_{(II)}$ $2p_{1/2}$, and Fe$_{(III)}$ $2p_{1/2}$, respectively, revealing that two kinds of iron ions (Fe$^{2+}$/Fe$^{3+}$) populate the $K_3Fe_2[PcFe–O_8]$ samples (Supplementary Fig. 5). The corresponding Fe$^{2+}$/Fe$^{3+}$ ratio was inferred to be ~2/1. The O ($1s$) spectra exhibit a Fe–O–C peak at a binding energy of 531.3 eV, further demonstrating the coordination reaction between O and Fe atoms. The signal referring to K in $K_3Fe_2[PcFe–O_8]$ XPS spectrum originates from the counter ions balancing the—otherwise—negatively charged system. Element analysis and TGA measurements (Supplementary Fig. 6) further define the chemical formula of the layered MOF as $K_3Fe_2[PcFe–O_8]\cdot 2.2H_2O$ ($C_{32}H_{12.4}Fe_3K_3N_8O_{10.2}$; named as $K_3Fe_2[PcFe–O_8]$ for short). In order to exclude the effect of water adsorption on the surface area and porosity, $K_3Fe_2[PcFe–O_8]$ was treated with supercritical carbon dioxide ($CO_2$) drying[29], after which low-pressure N$_2$ sorption was measured at 77 K. The obtained spectra (Supplementary Fig. 7) exhibit a superposition of type II and IVa isotherms[30] which is typical for nanosized porous crystals. The Brunauer-Emmett-Teller surface area of $K_3Fe_2[PcFe–O_8]$ is estimated as 206 m$^2$ g$^{-1}$, with the counter ions K$^+$ incorporated in the pores.

Powder X-ray diffraction (PXRD) measurements with Cu K$\alpha$ irradiation ($\lambda = 1.54$ Å) for $K_3Fe_2[PcFe–O_8]$ revealed a highly crystalline structure with prominent (100), (200), (300) and (001) peaks at $2\theta = 4.9°$, 9.8°, 14.7°, 26.6°, respectively (Fig. 1b and Supplementary Fig. 8). We combined DFT calculation to optimize the atomic structure of single layer MOF and then simulate different stacking arrangements of layer-stacked $K_3Fe_2[PcFe–O_8]$, e.g., AA, AA-inclined, AA-serrated, and AB stacking modes (Fig. 1b and Supplementary Fig. 9)[31]. The DFT results show that AA-serrated stacking geometry is energetically favored and its calculated PXRD pattern agrees well with the experimental result. The corresponding unit cell parameters with the $P1$ group are $a = b = 18.1$ Å, $\alpha = 89.5°$, $\beta = 89.8°$, and $\gamma = 89.3°$, indicating long-range ordering along the $ab$ plane. The weak and broad peak at 26.6° corresponding to the (001) facet suggests a 3.3 Å interval between layers in $c$-axis normal to the 2D plane. Additionally, the average size of crystalline domains in

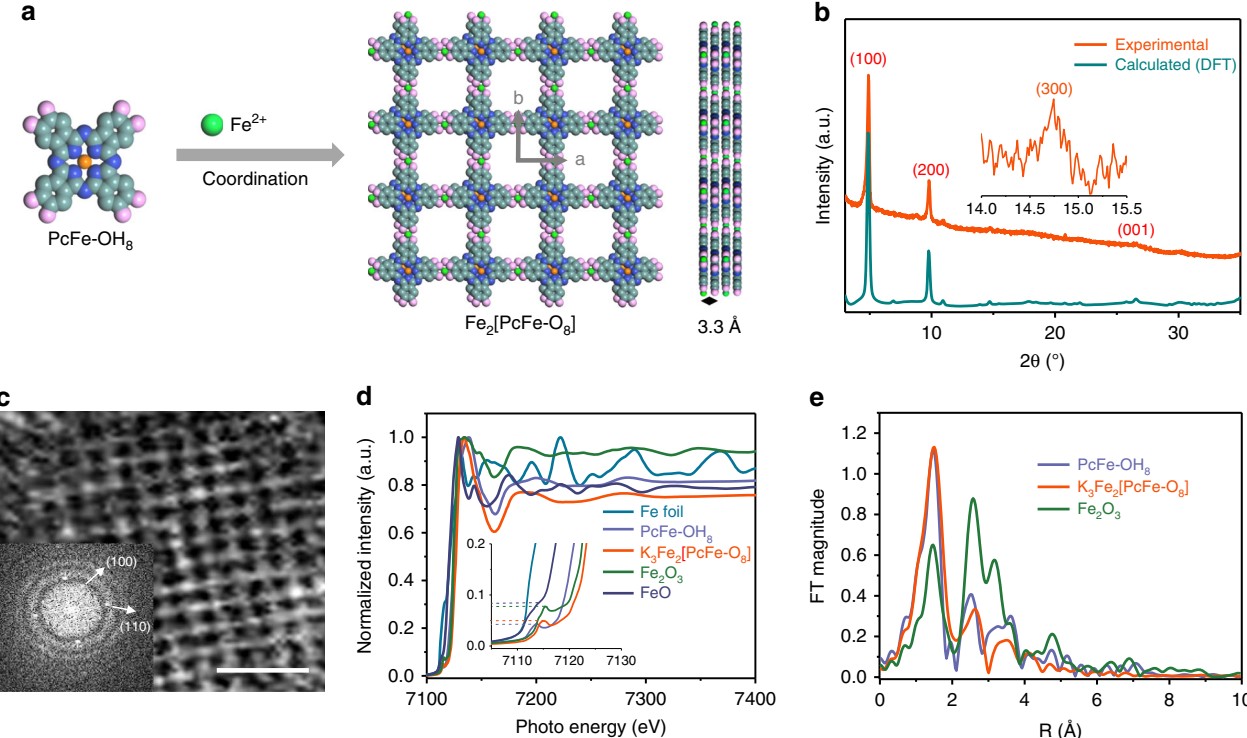

**Fig. 1** Synthesis and characterization of layered $K_3Fe_2[PcFe-O_8]$ MOF. **a** Schematic illustration for the synthesis of $Fe_2[PcFe-O_8]$ framework with iron ions and organic $PcFe-OH_8$ linkers connected by coordination bonds (light cyan: C; blue: N; light pink: O; orange: $Fe^{3+}$ in the phthalocyanine ring; green: $Fe^{2+}$ in the linkage; H atoms and $K^+$ counter-ions omitted for clarity). The interval between layers is about 3.3 Å along the c-axis; **b** Experimental PXRD pattern (organge curve) and that of calculated (DFT) AA-serrated stacking structures (dark cyan curve) of $K_3Fe_2[PcFe-O_8]$; **c** HRTEM image of $K_3Fe_2[PcFe-O_8]$. Scale bar: 5 nm. Inset: corresponding FFT analysis; **d** Normalized Fe K-edge XANES spectra of $K_3Fe_2[PcFe-O_8]$, Fe foil, FeO, $Fe_2O_3$, and $PcFe-OH_8$. Insert: enlarged pre-edge region in Fe K-edge XANES spectra; **e**, Fourier transformation EXAFS spectra at Fe K-edge of $Fe_2[PcFe-O_8]$ with $Fe_2O_3$ and $PcFe-OH_8$ as contrast

$K_3Fe_2[PcFe-O_8]$ was calculated to be ~36 nm according to the Scherrer's equation[32]. Transmission electron microscopy (TEM) results reveal rectangle-shaped crystals with sizes in the range of 10–100 nm (Supplementary Figs. 10 and 11). High-resolution TEM image (Fig. 1c) and the corresponding fast Fourier transform (FFT) image (inset Fig. 1c) manifest a $K_3Fe_2[PcFe-O_8]$ sample architecture based on a square lattice with a pore size of ~1.75 nm.

Synchrotron powder X-ray adsorption spectroscopy (XAS) was employed to further analysis the chemical state and coordination geometry in $K_3Fe_2[PcFe-O_8]$. Reference samples, e.g., Fe foil, FeO, $Fe_2O_3$ as well as the precursor $PcFe-OH_8$ were also investigated by XAS. Fig. 1d shows the X-ray adsorption near-edge structure (XANES) spectra at K edge of all the samples. The Fe K-edge of XANES in $K_3Fe_2[PcFe-O_8]$ exhibits a near-edge spectra similar to that of $PcFe-OH_8$ monomers, but completely different from those of FeO, $Fe_2O_3$, and Fe foil. Besides, the pre-edge feature (magnified in Fig. 1d) in $K_3Fe_2[PcFe-O_8]$ originated from the transition of 1s core electrons to hybridized orbitals of Fe (3d) and ligands (p)[33]. The intensity of the pre-edge peak is more intense on the site symmetry where the iron atom is located[34]. Figure 1e displays the Fourier transform of the $\kappa$-weighted extended X-ray absorption fine structure (EXAFS) of $K_3Fe_2[PcFe-O_8]$ as well as the contrast samples. The EXAFS presents a predominant peak in $K_3Fe_2[PcFe-O_8]$, which is originated from the nearest-neighboring nitrogen or oxygen coordination shell around the Fe atoms[35]. Based on this peak, Fe–N(O) distance was calculated to be ~1.57 Å. From the shape and amplitude of the main peak in the magnitude of the FT spectra, it is obvious that the bonding environment in

$K_3Fe_2[PcFe-O_8]$ is very close to that of the square planar geometry of $PcFe-OH_8$, further suggesting Fe atoms connected with four N/O atoms in the layered MOF. However, due to the limitation of the XAS resolution, we cannot distinguish the differences between $Fe-O_4$ and $Fe-N_4$ coordination geometries in the MOF. Nevertheless, another contrast sample $Fe_2O_3$ clearly exhibits two different predominant peaks at ~1.44 Å and ~2.57 Å, which arise from Fe–O and Fe–Fe bonds, respectively (Fig. 1e). Therefore, the XANES and EXAFS spectra of $K_3Fe_2[PcFe-O_8]$ and the contrast experiments provide strong proof on the formation of square planar complexes via the coordination of $PcFe-OH_8$ and Fe ions. Moreover, no metal oxides such as FeO and $Fe_2O_3$ were detected in the $K_3Fe_2[PcFe-O_8]$.

Compared with that of the constituent monomer $PcFe-OH_8$, the UV–Vis spectrum of $K_3Fe_2[PcFe-O_8]$ powders in N, N-Dimethylformamide (DMF) reveal a red-shift of the Soret-band to ~390 nm (Supplementary Fig. 12), testifying to the extended $\pi$-conjugation[36]. Cyclic voltammetry (CV) curves of $K_3Fe_2[PcFe-O_8]$ indicate a narrow energy gap of about 0.63 eV for the MOF powders dispersed in acetonitrile (Supplementary Fig. 13, Supplementary Table 3)[37].

**Electronic structure of $K_3Fe_2[PcFe-O_8]$.** To estimate the electronic structures of $K_3Fe_2[PcFe-O_8]$, we performed the DFT calculations. The band structure of a monolayer $K_3Fe_2[PcFe-O_8]$ with optimized geometry shows relatively dispersed bands on both sides of the Fermi level, suggesting a semiconducting behavior in this configuration with a rather narrow bandgap of ~0.12 eV (Fig. 2a, enlarged figure seen in Supplementary Fig. 14).

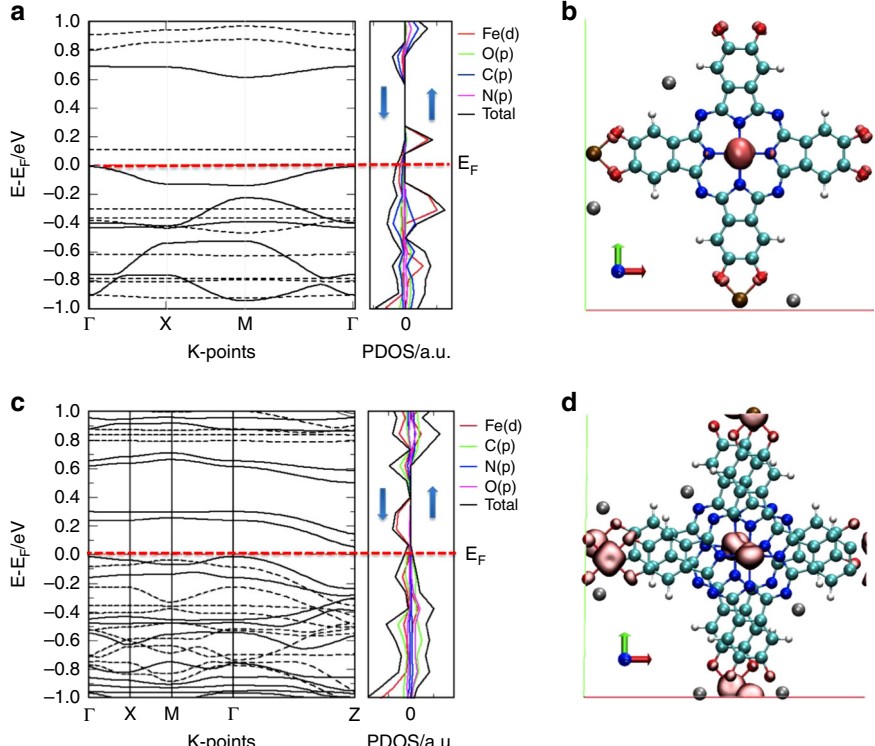

**Fig. 2** Modeling of the electronic structures of $K_3Fe_2[PcFe-O_8]$. **a** Band structure of a monolayer $K_3Fe_2[PcFe-O_8]$ with GGA + U correction shown on the left panel, dashed lines indicate the bands associated to spin up while the solid lines indicate the bands associated to spin down (the effective Coulomb ($U$) and exchange ($J$) terms reported in the Supplementary Information). The corresponding projected density of states (PDOS) for spin up and spin down are plotted on the right panel for Fe($d$), C($p$), O($p$), and N ($p$) states; **b** Spin density iso-surface (pink solid iso-surface), at absolute spin-density $|\rho \uparrow - \rho \downarrow| = 0.05$ electrons per $Å^3$ of a monolayer $K_3Fe_2[PcFe-O_8]$, indicating that the spin density is mainly localized on the Fe ions (light cyan: C; blue: N; red: O; light red: Fe; gray: $K^+$); **c** Calculated electronic band structure of multi-layered $K_3Fe_2[PcFe-O_8]$ with AA-serrated stacking mode; **d** Unit cell of two-layered $K_3Fe_2[PcFe-O_8]$ in AA-serrated stacking mode with ferromagnetic arrangement

The computational approach used in this study (DFT + U and PBE exchange-correlation functional, seen in Supplementary Methods), usually underestimates the bandgap in semiconductors[38]. Besides, the bands near enough to the Fermi level could be easily thermally-populated with holes, indicating a typical $p$-type semiconducting behavior. The PDOS near the Fermi level exhibits considerable hybridization of the orbitals from Fe($d$), C($p$), O($p$) and N($p$), which demonstrates a high degree of $\pi-d$ conjugation in the monolayer plane. The spin density iso-surface of $K_3Fe_2[PcFe-O_8]$ (Fig. 2b and Supplementary Fig. 15) reveals that the spin density is primarily localized on the Fe atoms in phthalocyanine cores, while the square planar Fe–$O_4$ moieties are slightly polarized by the polarization of the delocalized $\pi$ orbitals among Fe($d$), C($p$), O($p$) and N($p$)[28]. Notably, the oxidation state of Fe ions in the phthalocyanine ring is always higher than that in the Fe–$O_4$ moieties with their relatively higher Fe 2$p$ energy levels of 1.5–2.5 eV (Supplementary Fig. 16), which supports the coexistence of $Fe_{(II)}/Fe_{(III)}$ determined by XPS.

We further investigated the electronic band structure and PDOS of multi-layered $K_3Fe_2[PcFe-O_8]$. Fig. 2c displays a rather narrow band gap of ~0.10 eV in the AA-serrated stacking structure. In this case, Fig. 2d shows spin density iso-surfaces of two-layered $K_3Fe_2[PcFe-O_8]$ with unit-cell magnetization, in which the magnetic moments are also predominantly localized on the Fe atoms in phthalocyanine cores. Moreover, the magnetic ground state calculation of the AA-serrated stacking system reveals an exchange energy of $E_{ex} = 300$ meV ($E_{ex} = E_{AFM} - E_{FM}$), which implies the possible intrinsic ferromagnetic ordering of $K_3Fe_2[PcFe-O_8]$ (Supplementary Fig. 17).

**Charge transport and magnetotransport in $K_3Fe_2[PcFe-O_8]$.** The electrical conductivity of the $K_3Fe_2[PcFe-O_8]$ samples through *van der Pauw* method was derived to be $2 \times 10^{-3}$ S $m^{-1}$ at 350 K (measured in pellets with a thickness of ~0.59 mm, Supplementary Fig. 18).[25] Variable-temperature conductivity measurement presented a non-linear increase of electrical conductivity from 140 to 350 K, indicating a typical semiconducting behavior of the $K_3Fe_2[PcFe-O_8]$ sample (Fig. 3a and Supplementary Fig. 19). The electrical conductivity plotted versus reciprocal temperature ($T^{-1}$) indicates two thermally activated contributions to the conductivity of $K_3Fe_2[PcFe-O_8]$[39]. From the fitting results (inset in Fig. 3a), the activation energy in a high activation region was calculated to be ~261 meV, while that in a low-temperature activation region was ~115 meV (Supplementary Fig. 20). We ascribe this thermally-activated hopping progress to the grain boundaries dominating the temperature dependence of conductivity in the bulk polycrystalline pellets, thus giving rise to the relatively large activation energy[23,40]. In addition, it needs to mention that redox state of the iron centers also contributes to the conductivity of the $K_3Fe_2[PcFe-O_8]$. As contrast, after the sample was oxidized in the air for one week, the conductivity was 2 orders of magnitude lower than that before oxidation at room temperature (Supplementary Fig. 21a). The Fe ($2p$) core level XPS spectrum evidences the partial oxidation of $Fe^{2+}$ into $Fe^{3+}$ with a ratio of ~1/3 in the oxidized $K_3Fe_2[PcFe-O_8]$ (Supplementary Fig. 21b).

The Hall resistance ($R_{Hall}$) was further measured under magnetic field ($H$) based on the *van der Pauw* pattern (Supplementary Fig. 22), which displays a linear increase of $R_{Hall}$ with H. The fitting slope of the Hall effect shows a $p$-type

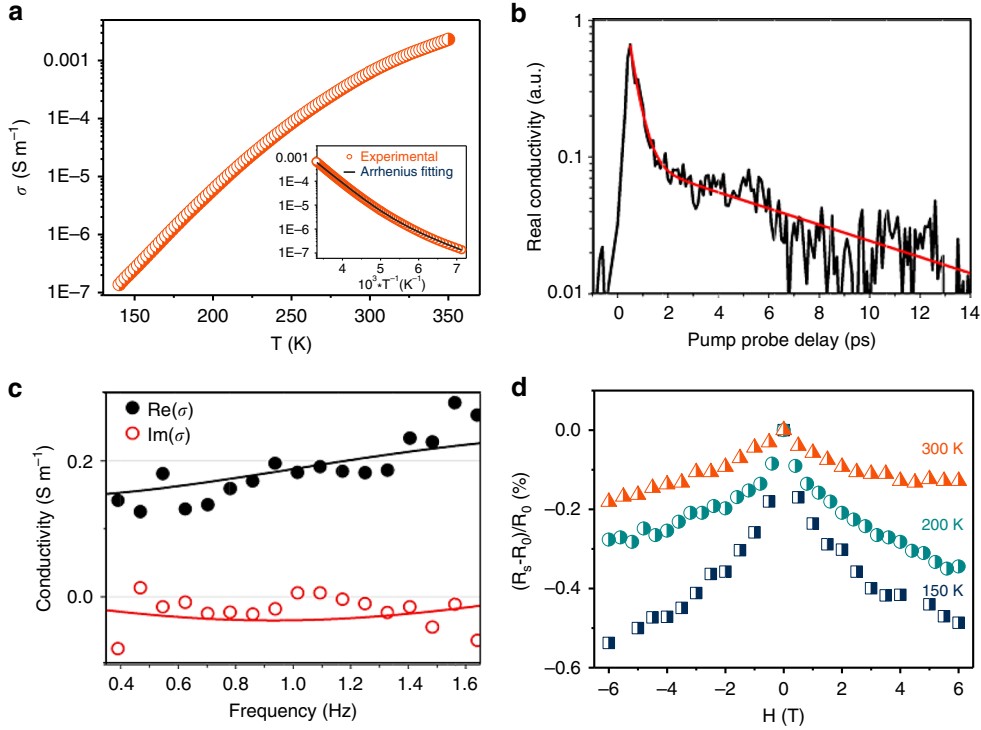

**Fig. 3** Charge transport and magnetoresistance of $K_3Fe_2[PcFe–O_8]$. **a** Variable-temperature electrical conductivity of compressed $K_3Fe_2[PcFe–O_8]$ pellets via *van der Pauw* method; Insert: Plot of electrical conductivity of $K_3Fe_2[PcFe–O_8]$ as a function of inverse temperature ($T^{-1}$); **b** Room-temperature photoconductivity of $K_3Fe_2[PcFe–O_8]$ measured by optical pump-THz probe spectroscopy (800 nm excitation wavelength 300 μJ cm$^{-2}$ and ~1 THz bandwidth probe; under nitrogen environment). Red line represents a bi-exponential decay function; **c** Real (black dots) and imaginary (red dots) components of the frequency-resolved complex conductivity (measured at 0.5 ps after photoexcitation, DC conductivity 0.14 S m$^{-1}$); solid lines represent a Drude-Smith description of the data; **d**, Magnetic field dependence of the magnetoresistance by measuring the changes of the electrical resistance in an applied field (−6-6 T) at different temperatures

semiconducting behavior in $K_3Fe_2[PcFe–O_8]$ with the hole concentration of about $2.4 \times 10^{14}$ cm$^{-3}$. The corresponding Hall mobility was calculated as ~0.1 cm$^2$ V$^{-1}$ s$^{-1}$. However, the carrier mobility from the Hall measurement is much lower than its intrinsic value, due to the contact resistance and the grain boundaries/gaps of $K_3Fe_2[PcFe–O_8]$ particles in the pressed pellets (Supplementary Fig. 23).

Next, contact-free time-resolved THz spectroscopy (TRTS) was employed to further address the photoconductivity and mobility nature in the $K_3Fe_2[PcFe–O_8]$ samples. Figure 3b shows the real part of the photoconductivity measured as a function of pump-probe delay for a ~210-micron thick sample (300 μJ cm$^{-2}$, 800 nm pump excitation, ~1 THz probe center frequency and ~1 THz bandwidth). These dynamics indicate the ultrafast (sub-*ps*) formation of quasi-free carriers that undergo rapid trapping and/or localization within the samples. Figure 3c shows the real and imaginary conductivity components as a function of frequency at the peak conductivity (0.5 ps after pump excitation); the data can be well described by the Drude-Smith model (shown as solid lines in Fig. 3c, seen in Supplementary Methods)[41]. This model for carrier transport is commonly employed to describe free charge carrier motion constrained by backscattering (as expected in our polycrystalline samples where long-range DC conductivity is suppressed by grain boundaries). The best fit to the data by the DS model provides a scattering time of 53 ± 8 fs and a *c* parameter of −0.69 ± 0.02 (where *c* ranges between 0 and −1; with 0 and −1 denoting completely free and completely localized *free* charges, respectively). From these estimates, and considering an *e-h* averaged effective mass of $m^*{\sim}1.9m_0$ (obtained from DFT calculations), we obtain a high mobility estimate for

the photo-generated free carriers at the peak conductivity of 15 ± 2 cm$^2$ V$^{-1}$ s$^{-1}$ (a mobility comparison with thus far reported stacked coordination polymers seen in Supplementary Table 4). Magnetotransport as a function of temperature was measured in the samples. Figure 3d reveals the change in resistance with varying applied magnetic field at 150, 200, and 300 K. As evident from the data, the resistance decreases with increasing magnetic field due to the reduced spin scattering[10], indicating the interaction between the magnetic moments and the carriers.

**Magnetic properties of the $K_3Fe_2[PcFe–O_8]$.** To probe the magnetic behavior of the $K_3Fe_2[PcFe–O_8]$, field- and temperature-dependent magnetization measurements were performed using superconducting quantum interference device (SQUID) magnetometry. Figure 4a shows the variation of the magnetization with the applied magnetic field (*M-H* curves) measured at different temperatures from 5 to 350 K (enlarged figure shown in Supplementary Fig. 24). Notably, at 350 K, $K_3Fe_2[PcFe–O_8]$ retains a magnetic hysteresis with a saturation magnetization of ~0.13 μB per Fe-site. Upon cooling to 5 K, an obvious increase in the coercivity and the magnetization was observed. Particularly, the hysteresis still did not reach the saturation at 7 T with magnetization of ~0.7 $\mu_B$ per Fe-site at 5 K. These observations suggest the superparamagnetic nature in the sample. Nevertheless, the magnetic hysteresis loops at 300 K and 350 K point to ferromagnetic coupling within single $K_3Fe_2[PcFe–O_8]$ crystallites with Curie temperature higher than 350 K. This is consistent with the DFT calculations which reveal an exchange energy of around 300 meV and also with the negative magnetoresistance at 300 K (Fig. 3d).

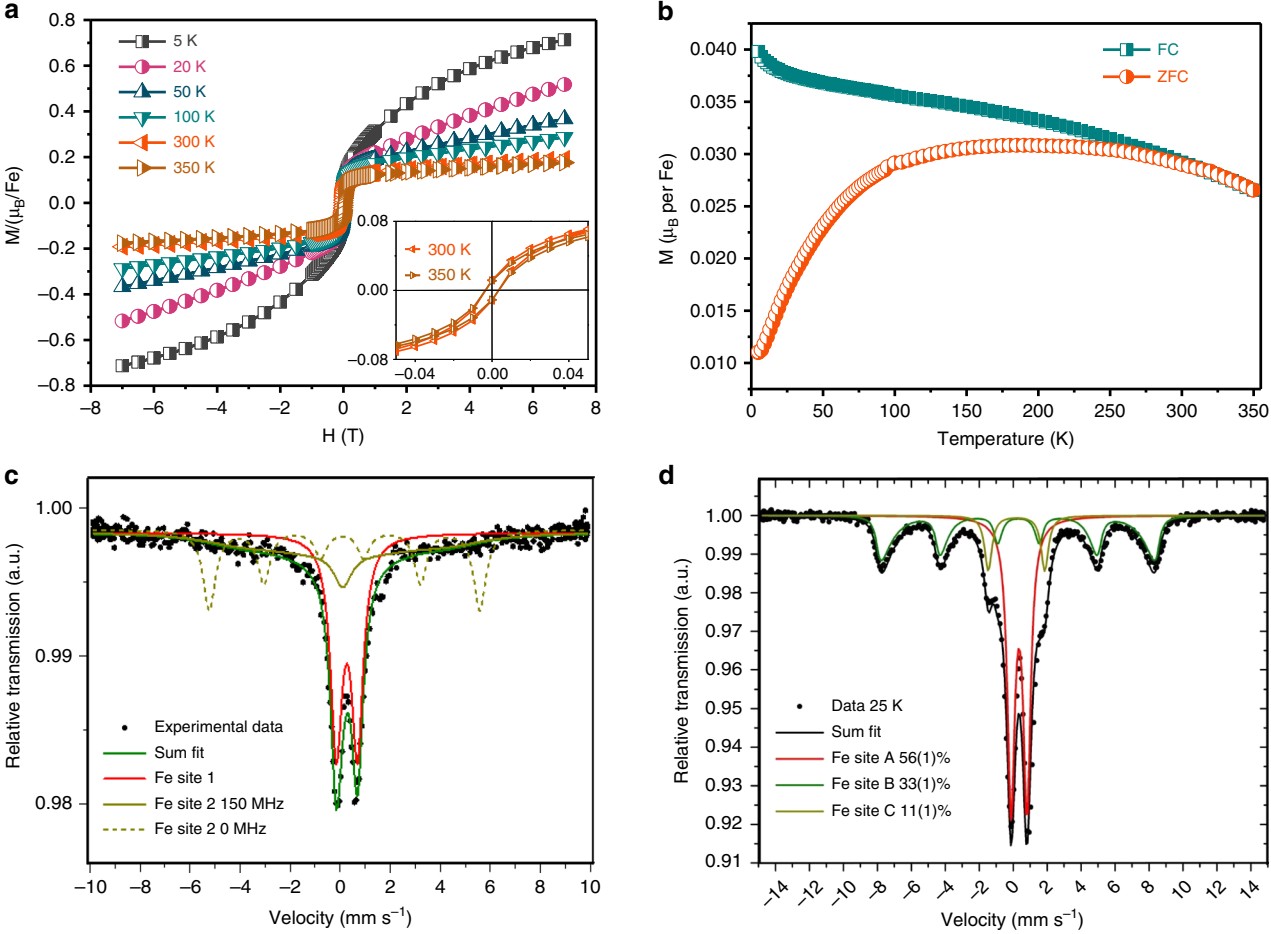

**Fig. 4** Magnetic properties of the $K_3Fe_2[PcFe–O_8]$. **a** Magnetic hysteresis loops obtained at different temperatures for $K_3Fe_2[PcFe–O_8]$; **b** Zero-field-cooled (ZFC) and field-cooled (FC) magnetization for $K_3Fe_2[PcFe–O_8]$ in an applied DC magnetic field of 100 Oe; **c**, **d** $^{57}Fe$ Mössbauer spectra of $K_3Fe_2[PcFe–O_8]$ measured at a temperature of 300 K and 25 K, respectively

The blocked superparamagnetic nature is also confirmed by the zero field cooled (ZFC) and field-cooled (FC) measurements (Fig. 4b). The FC magnetization in an external field of 100 Oe reveals a monotonic decrease upon increasing temperature. The corresponding ZFC magnetization at the same external field increases with temperature with a broad summit peak around 200 K, approaching the FC curve at around 300 K. The irreversibility between the ZFC and FC curves is due to blocked superparamagnetic clusters which have a broad distribution of blocking temperatures. This is the result of structural inhomogeneities in $K_3Fe_2[PcFe–O_8]$ (shown in the TEM images, Supplementary Figs. 10 and 11). The nanoscale $K_3Fe_2[PcFe–O_8]$ crystallites behave as single-domain superparamagnets. A splitting feature between the ZFC and FC curves is observed up to 300 K, which is also indicative of the persistence of magnetic couplings between the Fe spins in individual crystallites at or above 300 K. Notably, the remnant magnetization (Supplementary Fig. 25) does not decrease to zero at 350 K, further suggesting the presence of superparamagnetic particles with blocking temperatures even above 350 K. These results suggest that the short Fe–Fe distance in the $c$ direction together with the strong coupling between Fe spins mediated by the fully delocalized $\pi$-electrons along the $ab$ plane contributes to the persistence of spontaneous magnetization above room temperature. Compared with the Curie temperatures of thus far reported magnetic MOFs[14,18,42–44], our work discovers the layered conjugated MOF exhibiting ferromagnetic coupling above room temperature

(Supplementary Table 2). However, the polycrystalline nature of the samples (10–100 nm in domain size) renders the observed superparamagnetism, which could be addressed by improving the sample preparation, such as to obtain single crystals in the future.

## Discussion

To further understand the magnetic mechanism in $K_3Fe_2[PcFe–O_8]$, we investigated the local environment of Fe atoms by $^{56}Fe$ Mössbauer analysis. At 300 K, two kinds of Fe sites are resolved in the sample as shown in Fig. 4c. The narrow quadrupole doublet with an isomer shift of 0.378(4) mm s$^{-1}$ and an electric quadrupole splitting $\Delta E_Q = 0.843(6)$ mm s$^{-1}$ indicate the $Fe_{(III)}$-high spin oxidation state and provides the magnetic moments (Supplementary Table 5). The broad subspectrum assigned to Fe site 2 extends over the full velocity range with an isomer shift of 0.242(10) mm s$^{-1}$ and a slowly fluctuating magnetic hyperfine field (Blume model). The fit results in a magnetic hyperfine field of 30(5) T with a fluctuation frequency of 150(50) MHz. Such a subspectrum is often observed in nanoscale superparamagnetic particles above the blocking temperature. In contrast, the $^{57}Fe$ Mössbauer spectrum demonstrates the only presence of one $Fe_{(III)}$ state in the monomer PcFe–OH$_8$ (Supplementary Fig. 26), which reveals that phthalocyanine ligands contribute to partial Fe site 1 while the Fe site 2 originates from the linkages in $K_3Fe_2[PcFe–O_8]$ MOFs. This observation is consistent with the ferromagnetic exchange within the individual clusters[45].

Upon cooling from room temperature to 25 K, three Fe sites are observed in $K_3Fe_2[PcFe–O_8]$ (Fig. 4d) as expected. The spectrum at 25 K exhibit a sharp quadrupole splitting (Fe site A; isomer shift 0.35(1) mm s$^{-1}$; $\Delta E_Q = 0.94(1)$ mm s$^{-1}$), which is in consistent with a $Fe_{(III)}$ high spin states (Fig. 4d), while the Fe site C is in consistent with that observed in the monomer (Supplementary Fig. 26). Besides, Fe site B shows a magnetic hyperfine splitting, the sixtet of which is consistent with the subspectrum observed in ferromagnetic nanoparticles. The Mössbauer measurements suggest the superparamagnetic nature of the sample. In the superparamagnetic state, the magnetization direction of nanoparticle fluctuates among the easy axes of magnetization when there is no external magnetic field. The relaxation time depends on the size of the particles and the temperature. Therefore, magnetically, three sites are observed at 25 K because some particles are large enough to have a longer relaxation time, which contributes to the sixtet, while the remained randomly oriented particles are superparamagnetic (red site A).

Therefore, based on the $^{56}$Fe Mössbauer analysis as well as the support from the XPS (Supplementary Fig. 5c) and spin density distribution calculation (Fig. 2d), we can infer that the superparamagnetism in our $K_3Fe_2[PcFe–O_8]$ readily origins from the polycrystalline feature by regarding to the varied crystalline domain sizes. In addition, it is proposed that the magnetic coupling is induced by indirect exchange interaction between the localized iron spins via the highly delocalized $\pi$ electrons along the fully conjugated backbones[10,24], thus enabling strong hybridization between the $d/p$ orbitals of Fe, the phthalocyanine core, and the $Fe–O_4$ nodes (seen in Fig. 2c). As mentioned before, the calculated magnetic ground states of AA-serrated $K_3Fe_2[PcFe–O_8]$ present an energy-favorable ferromagnetic coupling with a positive exchange coupling energy of 300 meV (Supplementary Fig. 17), which further validates this conclusion. Nevertheless, it should be noted that further experiments are required to improve the crystalline quality to obtain ferromagnetic MOFs at or even above room temperature and sophisticated characterizations should be done to unambiguously exclude the possibility of magnetic secondary phases. Our work is expected to encourage more physical researches on magnetic and semiconducting properties of layered conjugated MOFs.

In summary, we have demonstrated a phthalocyanine-based MOF ($K_3Fe_2[PcFe–O_8]$) with layered structures, which exhibits a $p$-type semiconducting behavior with a high mobility of ~15 cm$^2$ V$^{-1}$ s$^{-1}$ at 300 K and blocked superparamagnetic nature up to 350 K (the highest temperature we have measured). DFT calculations indicate that the ferromagnetic ground state benefits from the strong hybridization between the $d$-$p$ orbitals of iron, the phthalocyanine core, and the iron-bis(dihydroxy) nodes. Our work highlights layered conjugated MOFs as a class of semiconducting materials for potential spintronics applications.

## Methods

**Materials**. All the starting materials were purchased from commercial suppliers, such as Sigma-Aldrich, TCI and abcr GmbH. Unless otherwise stated, all the chemicals were used directly without further purification. The reactions were performed with the standard vacuum-line and Schleck techniques under nitrogen. Colum chromatography was performed using the silica gel.

(2,3,9,10,16,17,23,24-octahydroxy phthalocyaninato)Fe (PcFe–OH$_8$) was synthesized according to the previous reported literature with minor modification (see Supplementary Information)[46].

All the reactions were performed under vacuum using the Schlenk line technique.

**Typical synthesis of $K_3Fe_2[PcFe–O_8]$**. In a 25 mL tube, PcFe-OH$_8$ (20 mg, 0.029 mmol), iron (II) acetate (9.4 mg, 0.054 mmol) were mixed together with KOAc (26 mg, 0.26 mmol) in N-methyl-2-pyrrolidone (NMP)/H$_2$O (3 mL/1 mL, v/v). The mixture was sonicated for 1 min, after which the system was degassed with freeze-pump-thaw methods for three times, sealed under high vacuum and kept at 423 K

(150 °C) for 3 days. Then, the mixture was cooled to room temperature, filtrated and soaked in degassed H$_2$O for 1 day. After washed by acetone and filtration, the layered $K_3Fe_2[PcFe–O_8]$ MOF was dried under vacuum at 40 °C as dark black powder. Yield: 88.4%. Element analysis and TGA measurements (Supplementary Fig. 6) define the chemical formula of the MOF as $K_3Fe_2[PcFe–O_8]\cdot 2.2H_2O$ ($C_{32}H_{12.4}Fe_3K_3N_8O_{10.2}$; found Fe, 17.3; C, 40.16; H, 1.38; N, 11.83; K, 12.13; O, 17.05; named as $K_3Fe_2[PcFe–O_8]$ for short).

**Variable-temperature conductivity measurements**. The pressed pellets were prepared by adding 25 mg samples (heated at 100 °C under vacuum overnight) onto a polymer film in an 8 mm inner diameter split sleeve pressing under the applied pressure of 1 GPa at 100 °C. After the temperature cooled down to room temperature, the pressed pellet was taken out and the thickness was measured. Then, four probes of silver wires were placed onto the top of the pressed pellets using conductive silver plastic. The probe was transferred onto the probe station with thermally conductive and electrically insulating grease. Then, the device was kept in air for 1 h to keep the complete drying of the paste (Supplementary Fig. 18). For all electrical measurements, standard sample resistance ranging from 0.04 mΩ to 200 GΩ is within the detectable limitation, and the temperature range is from 5 to 400 K by helium cooling. After confirming the Ohm contact at different temperatures, we collected an $I$-$V$ curve by scanning the current from 10 nA to 100 μA and measuring voltage at each step at every certain temperature. The electrical resistance is extracted from linear region of the $I$-$V$ curve. The average resistivity $\rho$ (Ω·m) of a sample is according to equation 1, where $R_s$ (Ω·square) is the sheet resistance and $t$ (m) is the thickness of sample. Hence the conductivity $\sigma$ (S m$^{-1}$) is calculated from Eq. 2.

$$\rho = R_s \cdot t \tag{1}$$

$$\sigma = \frac{1}{\rho} = \frac{1}{R_s \cdot t} \tag{2}$$

**Time-resolved THz spectroscopy**. The measurements were performed using an optical pump-THz probe setup driven by a titanium: sapphire laser amplifier system generating ~100 fs width laser pulses with a central wavelength of 800 nm, a pump fluence of 300 μJ cm$^{-2}$, and a repetition rate of 1 kHz. The ~210-micron thick sample (O.D. (800 nm) = 2.3) was sandwiched between fused silica substrates and measured in transmission under nitrogen environment. The phase-sensitive detection of the THz pulse allows the evaluation of the complex conductivity of the photoexcited sample[47]. The frequency-resolved complex photoconductivity spectra of the sample were fitted by using the Drude-Smith (DS) model (Eq. 3):

$$\sigma_{DS}(\omega) = \frac{\omega_p^2 \varepsilon_0 \, \tau}{1 - i\,\omega\,\tau} * \left(1 + \sum_{n=1}^{\infty} \frac{c_n}{(1 - i\,\tau)^n}\right) \tag{3}$$

where $\omega_p$, $\varepsilon_0$, and are the plasma frequency, vacuum permittivity, scattering time and backscattering parameter, respectively. Mobility estimates on the DC limit were obtained from the formula obtained from the formula $\sigma_{DS}(\omega \to 0) = eN\mu(1+c)$, where $\mu = e\frac{\tau}{m^*}$.

**Superconducting quantum interference device (SQUID)**. Magnetometry was performed by using a SQUID-VSM (Quantum Design). Temperature dependence of the magnetization of the layered MOF powder sample was measured in zero-field cooling (ZFC) and field cooling (FC) sequence with applied magnetic field of 100 Oe. The magnetic field dependence of magnetization was measured at different temperatures, i.e., 5, 10, 20, 50, 100, 300, and 350 K.

**$^{57}$Fe Mössbauer measurements**. Mössbauer measurements were carried out in a Cryo Vac helium flow cryostat with 6 litre helium volume protected by a nitrogen heat shield. We used commercial NIM rack devices. The drive was a Mössbauer WissEL drive unit MR-360 biased by a DFG-500 frequency generator in sinusoidal mode. We used for the data recording a CMTE multichannel data processor MCD 301/8 K and a WissEL single channel analyzer Timing SCA to select the energy window. The detector was a proportional counter tube and the source a Rh/Co source with an initial activity of 1.4 GB. The sample was a powder. The analysis was done by Moessfit[48].

**Modelling and electronic structure**. Density functional theory (DFT) calculations were carried out using the Vienna ab-initio Simulation Package (VASP)[49] version 5.4.1. The electronic wave-functions were expanded in a plane-wave basis set with a kinetic energy cutoff of 500 eV. Electron-ion interactions were described using the projector augmented wave (PAW) method[50]. Generalized gradient approximation (GGA)[51] of the exchange-correlation energy in the form of Perdew-Burke-Ernzerhoff (PBE)[52] was applied. We used DFT + U approach[53] to describe the localized d-orbitals of Fe ions. The effective Coulomb (U) and exchange (J) terms were set to 4 and 1 eV, respectively. Such combination of U and J were already successfully applied for very similar systems[28]. The MOF monolayer was modeled by adding a large vacuum space of 10 Å in the direction normal to the monolayer. Monkhorst-Pack[54] Gamma-centered grid with 6×6×1 dimension was used for k-

point sampling of the Brillouin zone. In the computational protocol for the 3D stacking of the studied MOF the k-point grid dimension was changed to $6 \times 6 \times 5$ and Grimme-D2 correction[55] was applied.

## Data availability

The data that support the findings of this study are available from the corresponding author on reasonable request.

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

## Acknowledgements

The authors thank financial support from ERC Grant on 2DMATER, and EU Graphene Flagship, Coordination Networks: Building Blocks for Functional Systems (SPP 1928, COORNET) as well as the German Science Council, Center of Advancing Electronics Dresden, EXC1056, (cfaed) and OR 349/1 and the Max Planck Society. The data of XANES and EXAFS were collected at room temperature in transmission mode at beamline BL14W1 and BL15U1 of the Shanghai Synchrotron Radiation Facility (SSRF, China). E.C. acknowledges financial support from the Max Planck Graduate Center and the regional government of Comunidad de Madrid under project (2017-T1/AMB-5207). We acknowledge Dresden Center for Nanoanalysis (DCN) at TUD, Dr. Petr Formanek and Dr. Konrad Schneider (Leibniz Institute for Polymer Research, IPF, Dresden) for the use of facilities. We also appreciate Prof. Bernd Büchner, Dr. Vladislav Kataev and Dr. Yulia Krupskaya (IFW Dresden) for the helpful discussion about the magnetic properties. T.H.and P.S.P.acknowledge the Centre for Information Services and HighPerformance Computing (ZIH) in Dresden, Germany for the provided computational resources.

## Author contributions

R.H.D. and X.L.F. designed the project. C.Q.Y., R.H.D., and M.C.W. synthesized the monomers. C.Q.Y. and R.H.D. synthesized the MOFs and performed the structural characterizations. C.Q.Y., R.H.D., Z.T.Z., and S.Q.Z. contributed to the SQUID measurements. C.Q.Y., M.W., R.H.D., M.C.W., and S.Q.Z. contributed to the variable-temperature conductivity together with the Hall effect measurements. P.S.P. and T.H. carried out the computational simulation and analysis. P.H., M. Ballabio., E.C., and M. Bonn contributed to the THz measurements and analysis. F.S. and S.K. contributes to the analysis of porous properties. J.C.Z. contributes to the XAS measurements. S.A.B. and H.-H.K. performed the Mössbauer measurements and analysis. Z.Q.L., R.H.D., and E.Z. carried out the HRTEM measurements. All the authors participated in discussing the data. R.H.D., X.L.F., and C.Q.Y. co-wrote the manuscript with contributions from all authors.

## Additional information

**Competing interests:** The authors declare no competing interests.

