## [Transparent Peer Review File · Nature Communications]

Reviewers' comments:

Reviewer #1 (Remarks to the Author):

The manuscript reports a phthalocyanine-based 2D MOF ($K_3Fe_2[PcFe-O_8]$) as a room-temperature ferromagnetic semiconductor with Curie temperature exceeding 350 K. Comprehensive experimental probes have been used to characterize the sample. The magnetization studies presented in Fig. 4 is convincing. The results are exciting, and the paper is generally well written. I would recommend it for publication in condition of the authors addressing the comments below.

- (1) Could the author clarify the physical properties difference between monolayer and very thick 2D MOF. It seems all measurements are done with very thick sample, i.e. 3D bulk samples, not really 2D.
- (2) Along the same lines, in the abstract and conclusion, where the characteristics of 2D MOF is stated, it will be good if the authors state clearly the thickness of the sample. Since all the characteristics are obtained from very thick samples (~1 mm or several hundred μm), not really at 2D limit.
- (3) Fig. 1a, could you add a zoom in plot of $Fe_2[PcFe-O_8]$ to show where the Fe^{2+} is located.
- (4) Fig. 3d and associated discussion in the main text. There are many possible causes for negative magnetoresistance. To avoid confusion, the author could consider remove the phrases "indicating the spin ordering of $K_3Fe_2[PcFe-O_8]$ after applying the magnetic field even at room temperature;" "": "The negative magnetoresistance indicates the coupling between the magnetism and the carriers in $K_3Fe_2[PcFe-O_8]$, thus revealing its intrinsic feature as a ferromagnetic semiconductor¹⁰."
- (5) Fig. 4b, how do you get remanent magnetization?
- (6) Fig. 4c shows the key evidence of magnetism. The presentation should be improved. Could you try waterfall plot with smaller symbol size? The inset is also hard to read.
- (7) Could you try Mössbauer spectroscopy at different temperature? It will be useful to understand the magnetic order.
- (8) Fig. 3. Charge transport: It will be beneficial for the readers if you could show $i-v$ curve at selected temperature (perhaps in supplementary material?)
- (9) Is it possible to obtain mobility from simple Hall measurement?
- (10) Line 245: why is the magnetization so small? "0.04 $\mu B/Fe$ -site at 5 K to 0.027 $\mu B/Fe$ -site at 350 K".
- (11) How thick is the sample for the measurement done in Fig. 4?

Reviewer #2 (Remarks to the Author):

The authors claim the discovery of a 2D MOF exhibiting room temperature ferromagnetism and semiconductivity. These are important claims of current interest in the field of functional MOFs. However, the reported results are puzzling and the conclusions not convincing and premature.

- 1) The authors use the term "2D material" in an incorrect manner. In the context that the community of 2D materials is using this term, which is the one used in the introduction (refs. 12 and 13), this term involves the isolation of an atomically-thin layer. However, the authors do not report the isolation of a 2D material. At best they can claim that they have reported a layered material !!! Even in this case, the proposed layered structure they claim is based on a very weak experimental support (PXRD data and local HRTEM) that does not allow to exclude other (non-crystalline) phases or (magnetic) impurities. The reported porosity seems to be too low and not fully consistent with the structural model.
- 2) It is interesting to observe that the material exhibits some electrical conductivity that allows to conclude that the material is a semiconductor, in agreement with the theoretical expectations.

However, the claim that the material orders ferromagnetically at room temperature (or even above) is not supported by the magnetic measurements. If the magnetic moments are predominantly localized in the high spin Fe(III) centers, much higher magnetic values have to be expected both in the temperature- and field-dependent magnetization measurements. The authors should check if this behavior is dominated by magnetic impurities (a 1% should be enough to account for the observed behavior). The absolute values of the magnetic moments seem in any case to be too low with respect to the expected magnetic moments coming from the Fe(III) magnetic centers. The fact that in the hysteresis loops the curves at different temperatures change (with an increase in the M vs H signal upon cooling down) also indicates that the compound is not a pure magnet, but contains, together with the magnetic contribution, a paramagnetic component which dominates at low temperatures. Mossbauer measurements at room temperature also suggest the presence of two different Fe(III) sites, one quadrupole doublet dominating the spectrum, which may come from isolated Fe(III), and a second very weak signal split by the hyperfine coupling, which should originate from these magnetic impurities.

3) Finally, well-characterized layered conductive magnets based on MOFs have been recently characterized (see *Nature Chem.* 10, 1056–1061 (2018)).

Reviewer #3 (Remarks to the Author):

This manuscript describes the synthesis and characterization of a new 2D metal-organic framework (MOF) that displays ferromagnetic behavior at room temperature. The discovery of such room-temperature ferromagnetics could potential enable alternative computing architectures to solve the Moore's Law problem based on spintronics. Many ferromagnetic materials are known, most of which are inorganic and that require cryogenic temperatures. Coordination polymers and in particular MOFs have recently become of interest due to the high degree of design flexibility they possess. Consequently, this report showing ferromagnetic behavior with $T_c > 300$ K is an important discovery. The manuscript describes extensive characterization of the material, including spectroscopic, magnetic, and charge transport manuscript. It is also well written and for the most part reasonably understandable to a general audience. I recommend publication assuming the following points are addressed:

- The material is not charge neutral and requires potassium cations for charge balancing. Presumably these are located within the MOF pores. Most likely because of this, the porosity of the material is not very high (75 m²/g, which is much lower than most MOFs). It would be interesting to know if the potassium ions could be exchanged for smaller Li⁺ or Na⁺ to increase the porosity.
- Related to this point is the limited information concerning the activation procedure used. The methods section (p. 17) says that the "mixture" was "filtrated and soaked in degassed H₂O for 1 day." By mixture, are the authors referring to a powder that formed and is suspended in the reaction solvent? Following the soaking, it is stated that the material was dried under vacuum. For how long was this done and at what temperature? If my presumption that this was done at room temperature is accurate, it seems likely that some water remains within the pores. The FTIR (Fig S2) suggests this is small, but Figure S5 shows an XPS peak assigned potentially to H₂O. Consequently, it would be good to rule out the effects of residual water by activating at a higher temperature (100 C is preferable. If the material won't tolerate that then at least something above 25 C) and then remeasuring the conductivity.
- In general, nothing is said about the potential role of water and the counterions in the conductivity. The conductivity of materials such as Prussian Blue and its analogues is well known to depend on water in the pores.
- Have the authors considered examining the effects of reduction or oxidation of the material as both a probe of the conductivity mechanism and as a potential strategy for tailoring the properties? Again referring to Prussian Blue, this is a mixed oxidation-state material with properties that must depend on the oxidation states of the two iron sites. Perhaps a simple experiment to test this idea could be added?

- The reasons for using MOFs as a strategy for creating new ferromagnetics are primarily the exceptional synthetic flexibility. However, the authors don't say anything about the potential advantages of having nanoporosity as well as the desired magnetic properties. Since the term MOF is synonymous with a porous material and will no doubt attract attention because of this, the authors should make some comment regarding potential advantages and uses of the porosity.
- Regarding prior reports of MOF ferromagnetics, I'm puzzled as to why the authors omit referencing their own 2018 paper in this journal describing a coronene-based MOF that is also ferromagnetic (R. Dong et al. NATURE COMMUNICATIONS | (2018)9:2637 | DOI: 10.1038/s41467-018-05141-4). Although the T_c of this material isn't reported for some reason, it must be at or below 20 K, so it doesn't compare with the material described in this manuscript. Nevertheless, it seems odd not to reference it in the intro and include it in Table S2.
- Figure 2 parts b and d: it is not easy to readily identify the spin isosurfaces because their color (white/gray) does not show up well against the white background. I suggest using a darker background or else different color scheme to make this easier. It would also be nice if the PDOS curves could be made larger so that it is easier to distinguish the colors of the various lines. Perhaps including larger versions in the SI would be an option.
- In figure 2a and c, what are the dashed black lines supposed to indicate? It also a bit unclear what is meant on p. 10 line 176 by the statement "as shown in Fig. 2a". this isn't obvious, so a little more explanation would be helpful.
- What is the explanation for the biexponential temperature dependence of the conductivity? The magnitude of the activation energy is attributed to grain boundaries, but why two different slopes?
- Page 13: the mobility value looks pretty good but it would be good to compare it with other 2D materials. Another interesting aspect is that the bandgap of this material is only 0.63 eV, whereas the inorganic materials in table S1 are all 1 eV or greater. Would this be an advantage for device applications?

Detailed response to the comments from the reviewers

Reviewer #1

General Comment: *The manuscript reports a phthalocyanine-based 2D MOF($K_3Fe_2[PcFe-O_8]$) as a room-temperature ferromagnetic semiconductor with curie temperature exceeding 350 K. Comprehensive experimental probes have been used to characterize the sample. The magnetization studies presented in Fig. 4 is convincing. The results are exciting, and the paper is generally well written. I would recommend it for publication in condition of the authors addressing the comments below.*

Response: We appreciate the Referee#1 for the encouraging comments and the positive recommendation for publication after revision. According to the reviewer's valuable suggestions, we have revised the manuscript to make our work more readable. Hope that our additional efforts in the revised manuscript address your concerns appropriately.

Comment 1. *Could the author clarify the physical properties difference between monolayer and very thick 2D MOF. It seems all measurements are done with very thick sample, i.e. 3D bulk samples, not really 2D.*

Response: Thanks for the insightful question with regards to the definition of "2D". From a broad definition, 2D MOFs can be regarded as atomically ordered metal-organic networks along 2D directions with strong in-plane bonding and weak out-plane bonding, which has been increasingly accepted by the MOF community (Nat. Mater. 2010, **9**, 565-571; Chem. Mater. 2012, **24**, 3511-3513; J. Am. Chem. Soc. 2013, **135**, 2462-2465; Angew. Chem. Int. Ed. 2015, **54**, 4349-4352; J. Am. Chem. Soc. 2017, **139**, 10863-10867; Nat. Energy 2018, **3**, 30-36; J. Am. Chem. Soc. 2018, **140**, 3040-3051; Nat. Mater. 2018, **17**, 1027-1032). Nevertheless, we agree with the referee that the terminology of "2D" given in our paper is a little misleading. Strictly speaking, $K_3Fe_2[PcFe-O_8]$ is indeed a bulk MOF with vdW layer-stacked structure. To avoid confusion, we have changed the '2D MOFs' into 'layered 2D MOFs' in the revised manuscript.

The physical properties are different between monolayer and multilayer MOFs in our system. For instance, as shown in Figure 2 (in the manuscript), the density state around Fermi level in monolayer is contributed from the hybridization of orbitals from Fe(*d*), C(*p*), O(*p*) and N(*p*), which means a high degree of π -*d* conjugation in plane. Due to the strong Fe(*d*) orbitals overlapping and enhanced *van der Waals* interactions between layers with AA-serrated stacking

mode, bulk layered $\text{K}_3\text{Fe}_2[\text{PcFe-O}_8]$ presents a different electronic structure with a narrowed band gap of ~ 0.10 eV. Therefore, the conductivity of layered MOF can be principally tuned by varying the layer numbers and the layer stacking geometry (Chem. Sci. 2017, **8**, 2859-2867), which is also unique challenge for the currently reported electroactive 2D materials (Science 2004, **306**, 666-669; Nat. Nanotech. 2012, **7**, 699-712; Nature 2017, **546**, 270-273).

Comment 2. Along the same lines, in the abstract and conclusion, where the characteristics of 2D MOF is stated, it will be good if the authors state clearly the thickness of the sample. Since all the characteristics are obtained from very thick samples (~ 1 mm or several hundred μm), not really at 2D limit.

Response: We are sorry for the inaccurate description of “2D” in the previous manuscript again. The variable-temperature conductivity and carrier mobility measurements were performed with the compressed pellets of $\text{K}_3\text{Fe}_2[\text{PcFe-O}_8]$ by a reported method (J. Am. Chem. Soc. 2017, **139**, 13608-13611; J. Am. Chem. Soc. 2018, **140**, 14533-14537). The thickness was measured as ~ 0.592 mm (given in the previous manuscript). The magnetic properties were investigated in its powder form.

Comment 3. Fig. 1a, could you add a zoom in plot of $\text{Fe}_2[\text{PcFe-O}_8]$ to show where the Fe^{2+} is located.

Response: Thanks for the constructive comment from the reviewer. To make the structural illustration of $\text{K}_3\text{Fe}_2[\text{PcFe-O}_8]$ more readable, we have utilized a more contrastive color for Fe^{2+} (as shown in Figure R1).

We have added this new figure in our revised manuscript (Figure 1a).

Figure R1 | Schematic illustration for the synthesis of $\text{Fe}_2[\text{PcFe-O}_8]$ framework with iron ions and organic PcFe-OH_8 linkers connected by coordination bonds (grey: C; blue: N; red:

O; light purple: Fe^{3+} ; green: Fe^{2+}). The interval between layers is about 3.3 Å along the *c*-axis.

Comment 4. Fig. 3d and associated discussion in the main text. There are many possible causes for negative magnetoresistance. To avoid confusion, the author could consider remove the phrases “indicating the spin ordering of $\text{K}_3\text{Fe}_2[\text{PcFe-O}_8]$ after applying the magnetic field even at room temperature;” “: “The negative magnetoresistance indicates the coupling between the magnetism and the carriers in $\text{K}_3\text{Fe}_2[\text{PcFe-O}_8]$, thus revealing its intrinsic feature as a ferromagnetic semiconductor¹⁰.”

Response: Thank you for your constructive comment on the negative magnetoresistance of $\text{K}_3\text{Fe}_2[\text{PcFe-O}_8]$. As the reviewer suggested, we have removed such inaccurate description in the revised manuscript.

Comment 5. Fig. 4b, how do you get remnant magnetization?

Response: In the revised manuscript, we have provided a detail about this measurement in Methods: “Under a magnetic field of 1000 Oe, the sample was cooled down from 350 K to 5 K sequentially. Then after removing the magnetic field, the remnant magnetic was measured as a function of temperature (from 5 K to 350 K) in zero field.”

Comment 6. Fig. 4c shows the key evidence of magnetism. The presentation should be improved. Could you try waterfall plot with smaller symbol size? The inset is also hard to read.

Response: Thanks for the kind suggestion from this reviewer. Figure R2 shows the waterfall plot of the enlarged magnetic hysteresis loops with the magnetic field obtained at different temperatures. We have added this figure in the SI (Supplementary Figure 25).

Figure R2 | Enlarged M-H curves of $\text{K}_3\text{Fe}_2[\text{PcFe-O}_8]$ at different temperatures showing hysteresis loops.

Comment 7. *Could you try Mössbauer spectroscopy at different temperature? It will be useful to understand the magnetic order.*

Response: As suggested by the reviewer, we have measured the ^{57}Fe Mössbauer spectroscopy at 25 K (Figure R3). Magnetically, three Fe sites are observed upon cooling from room temperature to low temperature. Since the isomer shift of 0.35(1) mm/s and the quadrupole splitting of $\Delta E_Q=0.94(1)$ mm/s of Fe site A is consistent with a Fe(III) High-Spin state (Figure 4d in the manuscript), Fe site C is consistent with the measured monomer (Supplementary Figure 19 in SI). Fe site B shows a magnetic hyperfine splitting. The magnetic behavior can be understood in the following way: the observed shape of the sextet is consistent with the subspectra observed in ferromagnetic nanoparticles, e.g. magnetic clusters. Probably, the ferromagnetic interaction depends on the size d (diameter) of these ferromagnetic nanoparticles and the sample contains a distribution of different sizes. For a critical diameter D of the nanoparticle the magnetic moments start to freeze at T_D and order. Therefore, magnetically, three sites are observed at 25 K because some nanoparticles are ordered while the remained randomly oriented nanoparticles are superparamagnetic (red site A).

We have added the figure and the corresponding discussion in the revised manuscript and SI (Supplementary Fig. 26b)

Figure R3 | ^{57}Fe Mössbauer spectra of $\text{K}_3\text{Fe}_2[\text{PcFe}-\text{O}_8]$ measured at a temperature of 25 K.

Comment 8. *Fig. 3. Charge transport: It will be beneficial for the readers if you could show I - V curve at selected temperature (perhaps in supplementary material?).*

Response : We do agree with the reviewer that the I - V curves should be included in the transport

properties, as shown in Figure R4. In fact, we have provided the I - V curves at different temperatures in Supplementary Figure 20 (in SI) in our previous version.

Figure R4 | I - V curves for $K_3Fe_2[PcFe-O_8]$ collected at 140 K, 200 K and 250 K display Ohmic responses.

Comment 9. Is it possible to obtain mobility from simple Hall measurement?

Response: We have tried to perform preliminary Hall measurement by the *van der Pauw* geometry (Philips Res. Rep. 1958, **13**, 1-9; Philips Tech. Rev. 1958, **20**, 220-224) on the compressed pellet samples. Based on the basic physical principle of the Hall effect, the carrier mobility can be estimated using standard techniques under a magnetic field B ranging from -6 T to 6 T.

Figure R5 | Magnetic field dependence of the Hall resistance (R_{Hall}) at 300 K, indicating the p -type semiconductor of $K_3Fe_2[PcFe-O_8]$ (H : applied magnetic field). Dot: experimental results; Line: fitting results.

As shown in Figure R5, the fitting slope of the Hall effect shows a p -type semiconducting behavior in $K_3Fe_2[PcFe-O_8]$, with the hole concentration of about $2.4 \times 10^{14} \text{ cm}^{-3}$ (calculated via $n_s = \frac{IB}{eV_H}$, where I is the current, B is the magnetic field, V_H is the Hall voltage and e is the

elementary charge). The corresponding mobility is about $0.1 \text{ cm}^2/\text{Vs}$, where the carrier mobility is calculated through $\mu = \frac{1}{en_s R_s}$. However, the carrier mobility from the Hall measurement is much lower than its intrinsic value, due to the contact resistance and the grain boundaries/gaps of $\text{K}_3\text{Fe}_2[\text{PcFe-O}_8]$ particles in the pressed pellets. To overcome the above disadvantages, time-resolved THz spectroscopy (TRTS) was employed to estimate the nature of charge mobility without the contact resistance, which provided a mobility as high as $15 \pm 2 \text{ cm}^2/\text{Vs}$.

We have added this figure and the corresponding discussion in the revised manuscript and SI (Supplementary Fig. 23).

Comment 10. Line 245: why is the magnetization so small? “ $0.04 \mu\text{B}/\text{Fe-site}$ at 5 K to $0.027 \mu\text{B}/\text{Fe-site}$ at 350 K”.

Response: We fully appreciate the keen comments from this reviewer. The small magnetization ($0.04 \mu\text{B}/\text{Fe-site}$ at 5 K to $0.027 \mu\text{B}/\text{Fe-site}$ at 350 K) was obtained from the ZFC/FC results in Figure 4a (in the manuscript). However, the ZFC/FC magnetization was measured at a limited applied magnetic field of 100 Oe, which is usually applied to investigate the temperature dependence of the magnetic properties (Nat. Commun. 2018, **9**, 2637-2645). Under such low magnetic field, the Fe spins are far from being fully polarized and thus the not-yet-saturated magnetization of $\text{K}_3\text{Fe}_2[\text{PcFe-O}_8]$ at 100 Oe cannot reflect the magnetic moment per Fe-site.

In order to estimate the magnetic moment per Fe-site from the macroscopic magnetic property (i.e. the saturation magnetization), the Fe spins shall be fully polarized along the applied magnetic field, which requires strong enough magnetic field and low enough temperature. In the present work, the strongest magnetic field and lowest temperature used were 7 T (namely 70000 Oe) and 5 K respectively (as the M - H measurement shown in Figure 4c). At this condition, the magnetization of $\text{K}_3\text{Fe}_2[\text{PcFe-O}_8]$ is around $0.7 \mu\text{B}/\text{Fe}$, almost 18 times larger than that measured under 100 Oe. Even so, we noticed that the magnetization still did not reach its saturated state, due to the existence of superparamagnetic particles which consist of small ferromagnetic nanoscale clusters. Therefore, the magnetic moment per Fe-site should be larger than $0.71 \mu\text{B}/\text{Fe}$.

Comment 11. How thick is the sample for the measurement done in Fig. 4?

Response: The magnetic characterization of $\text{K}_3\text{Fe}_2[\text{PcFe-O}_8]$ was performed on polycrystalline

powder-samples sealed in a polytetrafluoroethylene bag with the weight of 5.6 mg, which has been added in the revised manuscript.

Reviewer #2

General Comment: The authors claim the discovery of a 2D MOF exhibiting room temperature ferromagnetism and semiconductivity. These are important claims of current interest in the field of functional MOFs. However, the reported results are puzzling and the conclusions not convincing and premature.

Response : We fully appreciate the Reviewer #2 for his/her critical scientific comments, which drives us to further analyze the structure/composition and the magnetic properties in the current work, thus building up a reliable structure-property relationship. And we feel very sorry for such premature discussion in the previous version. To further address the reviewer's concerns, we have remeasured the surface area after further treatment by supercritical CO₂ drying of the samples and performed additional structural/compositional characterizations to exclude the influence of possible impurities in our samples including X-ray absorption near edge structure (XANES) spectra, extended X-ray adsorption fine structure (EXAFS) spectra, HR-TEM and elemental analysis. We also double checked the magnetic properties of different batches of MOF samples by SQUID and provided more understanding about the magnetic mechanism in our revised manuscript. We hope our additional efforts appropriately address your concerns.

Comment 1. The authors use the term "2D material" in an incorrect manner. In the context that the community of 2D materials is using this term, which is the one used in the introduction (refs. 12 and 13), this term involves the isolation of an atomically-thin layer. However, the authors do not report the isolation of a 2D material. At best they can claim that they have reported a layered material!!! Even in this case, the proposed layered structure they claim is based on a very weak experimental support (PXRD data and local HRTEM) that does not allow to exclude other (non-crystalline) phases or (magnetic) impurities. The reported porosity seems to be too low and not fully consistent with the structural model.

Response: We fully appreciate the insightful suggestion on the definition of "2D Materials" and the critical comment on non-crystalline phase or impurities. From a broad definition, 2D MOFs can be regarded as atomically ordered metal-organic networks along 2D directions with strong in-plane bonding and weak out-plane bonding, which has been increasingly accepted by the MOF community (Nat. Mater. 2010, **9**, 565-571; Chem. Mater. 2012, **24**, 3511-3513; J. Am. Chem. Soc. 2013, **135**, 2462-2465; Angew. Chem. Int. Ed. 2015, **54**, 4349-4352; J. Am. Chem. Soc. 2017, **139**, 10863-10867; Nat. Energy 2018, **3**, 30-36; J. Am. Chem. Soc. 2018, **140**, 3040-3051; Nat. Mater.

2018, **17**, 1027-1032). We agree with the Referee that the terminology of “2D” given in our paper is misleading. The synthetic $K_3Fe_2[PcFe-O_8]$ is indeed a bulk MOF with vdW layer-stacked structure. To avoid the confusion, we have changed “2D MOFs” into “layered 2D MOFs” in our revised manuscript.

We also fully understand the concerns from the reviewer on the crystallinity and the structural determination (such as repeated units, stacking pattern, amorphous structures and impurities), which are also the key research motivation of our group towards the chemistry of the controlled synthesis. As one of the common targets, it remains challenging for the community to synthesize highly crystalline or even single-crystalline layer-stacked 2D MOFs and achieve a strongly reliable structure-property relationship. Notably, a few successful examples about the magnetic layered MOFs have been recently reported by Coronado et al. (Nat. Chem. 2018, **10**, 1001-1007 (Published online: 27 August 2018)), Clérac et al. (Nat. Chem. 2018, **10**, 1056-1061(Published online: 10 September 2018)), Harris et al. (J. Am. Chem. Soc. 2015, **137**, 15699-15702; J. Am. Chem. Soc. 2017, **139**, 4175-4184) and Long et al. (J. Am. Chem. Soc. 2015, **137**, 15703-15711; J. Am. Chem. Soc. 2018, **140**, 3040-3051). Here, we submitted our work (Submission date: 28 August 2018) about the synthesis of a magnetic, polycrystalline, layer-stacked 2D MOF, which comprises of poly-dispersed nanoscale crystallites. To further address your concerns on possible impurities and surface area in layered MOF, we have carried out additional compositional and porous structural analysis by combining with the successful experience from the above published reports.

1. We excluded the possible impact of oxides impurities in $K_3Fe_2[PcFe-O_8]$ on the magnetic behavior within the current analytical resolution.

Synchrotron powder X-ray adsorption spectroscopy (XAS) was employed to further analyze the chemical state and coordination geometry in $K_3Fe_2[PcFe-O_8]$. Reference samples, e.g. Fe foil, FeO, Fe_2O_3 as well as the precursor PcFe-OH₈ were also investigated by XAS. Figure R6a shows the X-ray adsorption near-edge structure (XANES) spectra at K edge of all the samples. The Fe K-edge of XANES in $K_3Fe_2[PcFe-O_8]$ exhibits a near-edge spectra similar to that of PcFe-OH₈ monomers, but completely different from those of FeO, Fe_2O_3 and Fe foil. Besides, the pre-edge feature (magnified in Figure R6b) in $K_3Fe_2[PcFe-O_8]$ originated from the transition of 1s core

electrons to hybridized orbitals of Fe ($3d$) and ligands (p). The intensity of the pre-edge peak is more intense on the site symmetry where the iron atom is located (Nat. Commun. 2018, **9**, 935-747). Figure R6c displays the Fourier transform of the κ -weighted extended X-ray absorption fine structure (EXAFS) of $\text{K}_3\text{Fe}_2[\text{PcFe-O}_8]$ as well as the contrast samples. The EXAFS presents a predominant peak in $\text{K}_3\text{Fe}_2[\text{PcFe-O}_8]$, which originates from the nearest-neighboring nitrogen or oxygen coordination shell around the Fe atoms (Energy Environ. Sci. 2018, **11**, 22208-2215). Based on this peak, Fe-N(O) distance is calculated to be $\sim 1.57 \text{ \AA}$. From the shape and amplitude of the main peak in the magnitude of the FT spectra, it is obvious that the bonding environment in $\text{K}_3\text{Fe}_2[\text{PcFe-O}_8]$ is very close to that of the square planar geometry of PcFe-OH_8 , further suggesting Fe atoms connected with four N/O atoms in the layered MOF. However, due to the limitation of the XAS resolution, we cannot distinguish the differences between Fe- O_4 and Fe- N_4 coordination geometries in the MOF. Nevertheless, another contrast sample Fe_2O_3 clearly exhibits two different predominant peaks at $\sim 1.44 \text{ \AA}$ and $\sim 2.57 \text{ \AA}$, which arise from Fe-O and Fe-Fe bonds, respectively. Therefore, the XANES and EXAFS spectra of $\text{K}_3\text{Fe}_2[\text{PcFe-O}_8]$ and the contrast experiments provide strong proof on the formation of square planar complexes via the coordination of PcFe-OH_8 and Fe ions. Moreover, no metal oxides such as FeO and Fe_2O_3 were detected in the $\text{K}_3\text{Fe}_2[\text{PcFe-O}_8]$.

Figure R6 | Synchrotron X-ray adsorption spectroscopy of $\text{K}_3\text{Fe}_2[\text{PcFe-O}_8]$. a, Normalized Fe K-edge XANES spectra of $\text{K}_3\text{Fe}_2[\text{PcFe-O}_8]$, Fe foil, Fe_2O_3 and PcFe-OH_8 ; b, Expanded pre-edge region in Fe K-edge XANES spectra for $\text{K}_3\text{Fe}_2[\text{PcFe-O}_8]$, Fe foil, FeO, Fe_2O_3 and PcFe-OH_8 ; c, Fourier transformation EXAFS spectra at Fe K-edge of $\text{K}_3\text{Fe}_2[\text{PcFe-O}_8]$ with Fe_2O_3 and PcFe-OH_8 as the contrast samples;

^{57}Fe Mössbauer spectroscopy (Figures R7, Figure 4d in manuscript), which is also very sensitive to the local environment of Fe atom and its valence, displays that two kinds of Fe sites exist in the MOF sample at 300 K. The narrow quadrupole doublet with an isomer shift of 0.378(4)

mm/s and an electric quadrupole splitting $\Delta E_Q = 0.843(6)$ mm/s indicate the Fe(III)-high spin oxidation state and provides the magnetic moments. The broad subspectrum assigned to Fe site 2 extends over the full velocity range with an isomer shift of 0.242(10) mm/s and a slowly fluctuating magnetic hyperfine field (Blume model). The fit results in a magnetic hyperfine field of 30(5) T with a fluctuation frequency of 150(50) MHz. Such a subspectrum is often observed in nanoscale superparamagnetic particles above the blocking temperature. In contrast, the ^{57}Fe Mössbauer spectrum demonstrates the only presence of one Fe(III) state in the monomer PcFe-OH_8 (Supplementary Fig. 26a), which reveals that phthalocyanine ligands contribute to partial Fe site 1 while the Fe site 2 originates from the linkages in $\text{K}_3\text{Fe}_2[\text{PcFe-O}_8]$ MOFs. This observation is consistent with the ferromagnetic exchange interactions within the individual clusters. Thus, we can also exclude the iron oxides component in our system from the Mössbauer spectrum, due to that the harder iron oxides should be more strongly magnetically ordered than that of softer metal-organic components. Therefore, Mössbauer spectra also confirm no iron oxides purities in the $\text{K}_3\text{Fe}_2[\text{PcFe-O}_8]$.

Figure R7 | ^{56}Fe Mössbauer spectra at 300 K. Dots correspond to the experimental data and the solid lines represent the fitting spectra.

Following the suggestion from the reviewer, the TGA and the elemental analysis of $\text{K}_3\text{Fe}_2[\text{PcFe-O}_8]$ were performed to investigate the exact composition of $\text{K}_3\text{Fe}_2[\text{PcFe-O}_8]$. As shown in Figure R8, the TGA spectrum shows that the $\text{K}_3\text{Fe}_2[\text{PcFe-O}_8]$ starts to desolvate H_2O at 353 K (80 °C). A weight loss of 4.11% is an evidence of the coordinated water in $\text{K}_3\text{Fe}_2[\text{PcFe-O}_8]$. This result implies the existence of ~ 2 coordination H_2O , leading to a chemical formula for this synthetic layered MOF as $\text{K}_3\text{Fe}_2[\text{PcFe-O}_8]\cdot 2\text{H}_2\text{O}$. In our previous version, the high-resolution O

2p XPS spectrum (Supplementary Figure 5d) also shows the H₂O peak at 533.5 eV (Chem. Soc. Rev. 2013, **42**, 5833-5857), which is in consistency with the TGA results. In order to check the validity of our K₃Fe₂[PcFe-O₈]•2H₂O formula, we have performed elemental analysis by a combination of inductively coupled plasma optical emission spectrometry (ICP-OES) and C, H, O and N combustion method; the results (in percentage) are presented in the table below:

	Fe(%)	C(%)	H(%)	N(%)	K(%)	O(%)
Found	17.39	40.16	1.38	11.83	12.13	17.05
Calculated	17.57	40.32	1.27	11.75	12.30	16.78

From these estimates, and taking into account of the formula C₃₂H₁₂Fe₃K₃N₈O₁₀, we can obtain the following elemental ratios: C/Fe ≈ 32/3; C/H ~ 32/13.2; C/N ≈ 32/8; C/K ≈ 32/3; C/O ≈ 32/10.2. Therefore, the element analysis defines a chemical formula as K₃Fe₂[PcFe-O₈]•2.2H₂O (C₃₂H_{12.4}Fe₃K₃N₈O_{10.2}) for the layered 2D MOF that we have developed (named as K₃Fe₂[PcFe-O₈] for short).

Figure R8 | TGA analysis of K₃Fe₂[PcFe-O₈] under nitrogen atmosphere, indicating the existence of coordination H₂O in the layered MOF.

Regarding to the concern from the reviewer on the possible non-crystalline phase in the polycrystalline layered 2D MOFs, to be frank, we agree with the referee that it is challenging to exclude such possibility from the current XRD (Figure 1b in the manuscript) and local HR-TEM (Figure 1d) results. Nevertheless, we put our effort to exclude the possible impurities in the resultant K₃Fe₂[PcFe-O₈] by the above spectroscopy methods. In order to further address the referee's

concern, we have acquired a number of TEM images within different regions of the powder sample, as shown in Figure R9. Based on these TEM images, the rectangle-shaped nanocrystals of $\text{K}_3\text{Fe}_2[\text{PcFe-O}_8]$ are well visualized. We did not observe the existence of the amorphous phases throughout the sample. In addition, it is always challenging for the TEM community to acquire high-resolution structural image of 2D MOFs, due to their high sensitivity to electron beam (Science 2018, **359**, 675-679; Nat. Mater. 2017, **16**, 532-536; J. Am. Chem. Soc. 2017, **139**, 19-22; Angew. Chem. Int. Ed. 2015, **54**, 12058-12063). In fact, our work indeed offers a high-quality structural resolution by TEM (Figure 1c in the main text), which clearly presents nanoscale crystalline domains in $\text{K}_3\text{Fe}_2[\text{PcFe-O}_8]$ comprising square-patterned structure.

Figure R9 | TEM images of $\text{K}_3\text{Fe}_2[\text{PcFe-O}_8]$ within different regions of the powder samples.

2. Porous property was re-checked by treating the sample with supercritical CO_2 drying.

Thank the referee for pointing out the question about the low porosity. Here, we re-measured the N_2 sorption after further activation of the powder sample. Before the BET measurement, the powder sample was treated in a supercritical CO_2 dryer for 5 days, and then activated at 80°C overnight. As shown in Figure R10a (red), the newly resultant BET surface area of $\text{K}_3\text{Fe}_2[\text{PcFe-O}_8]$

is about 206 m²/g. A pore distribution curve shows the average microporous size as ~ 1.4 nm, which is smaller than the calculated value (~ 1.8 nm), due to the incorporation of K⁺ ions inside the pores (Figure R10b, red line). The larger pore sizes of 25 nm and 35 nm, respectively, estimated from the pore distribution originate from the aggregates of the nanoparticles (Supplementary Fig. 3 in SI). As contrast, after we changed the K⁺ ions by smaller Li⁺ ions (Li_xFe₂[PcFe-O₈]), the synthesis was performed according to the same method for K₃Fe₂[PcFe-O₈]), the BET surface area for Li_xFe₂[PcFe-O₈] was enhanced to be 343 m²/g (Figure R10a, blue line) and the average microporous pore size was determined as ~1.45 nm (Figure R10b, blue line), which further revealed the blocking influence of the incorporated counter ions on the porosity.

Figure R10 | Porous analysis of K₃Fe₂[PcFe-O₈] (red curves) and Li_xFe₂[PcFe-O₈] (blue curves). (a) N₂ adsorption isotherm measured at 77 K after supercritical CO₂ drying. (b) Pore size distribution profile calculated from the adsorption data using the DFT model. The synthesis of Li_xFe₂[PcFe-O₈] was performed according to the same method for K₃Fe₂[PcFe-O₈].

ACTIONS

- 1) We have revised the “2D MOF” as “layered 2D MOF” in our revision.
- 2) We have provided additional compositional analysis in the revised manuscript and SI, such as XANES (Fig. 1d), EXAFS (Fig. 1e), TGA analysis (Supplementary Fig. 6), element analysis (SI) and HR-TEM images (Supplementary Fig. 10).
- 3) We have re-measured the BET by treating the sample with supercritical CO₂ drying. The corresponding figures have been added as Supplementary Fig. 7 in the revised SI.

Comment 2. It is interesting to observe that the material exhibits some electrical conductivity that

allows to conclude that the material is a semiconductor, in agreement with the theoretical expectations. However, the claim that the material orders ferromagnetically at room temperature (or even above) is not supported by the magnetic measurements. If the magnetic moments are predominantly localized in the high spin Fe(III) centers, much higher magnetic values have to be expected both in the temperature- and field-dependent magnetization measurements. The authors should check if this behavior is dominated by magnetic impurities (a 1% should be enough to account for the observed behavior). The absolute values of the magnetic moments seem in any case to be too low with respect to the expected magnetic moments coming from the Fe(III) magnetic centers. The fact that in the hysteresis loops the curves at different temperatures change (with an increase in the M vs H signal upon cooling down) also indicates that the compound is not a pure magnet, but contains, together with the magnetic contribution, a paramagnetic component which dominates at low temperatures. Mossbauer measurements at room temperature also suggest the presence of two different Fe(III) sites, one quadrupole doublet dominating the spectrum, which may come from isolated Fe(III), and a second very weak signal split by the hyperfine coupling, which should originate from these magnetic impurities.

Response: We are grateful to the reviewer for pointing this out. Here, we would like to response your questions and comments point by point.

1. First of all, as discussed in the response to **Comment 1**, we can exclude the possible influence of magnetic impurities, such as FeO and Fe₂O₃, by our data from XPS, XANES, EXAFS, ⁵⁶Fe Mössbauer spectra, TGA, elemental analysis and HR-TEM. From a solid-state perspective, the possibility of having a concentration of magnetic impurities on the order of the 1% could be considered as strong doping in the samples. We believe that by providing a more physically meaningful quantification given by the setup detection limit, we are simply quantifying that the MOF sample dominates the magnetic behavior.

2. Regarding to the Fe oxidation state, we would like to clarify that there are indeed two kinds of Fe in K₃Fe₂[PcFe-O₈], i.e., Fe³⁺ and Fe²⁺, which makes this system more complicated. As shown in Supplementary Figure 5c (in the SI), the core level spectrum of Fe (2p) contains peaks at 710.8, 713.4, 724.2, 726.6 eV, attributable to the Fe_(II) 2p_{3/2}, Fe_(III) 2p_{3/2}, Fe_(II) 2p_{1/2}, and Fe_(III) 2p_{1/2}, respectively, revealing two kinds of iron ions (Fe²⁺/Fe³⁺). The corresponding Fe²⁺/Fe³⁺ ratio was inferred to be ~2/1. The ⁵⁷Fe Mössbauer spectrum of K₃Fe₂[PcFe-O₈] at 300 K also demonstrates two iron sites (Figure R7) and reveals that phthalocyanine ligands contribute to Fe³⁺ while the Fe²⁺ originates from the linkages after comparison with that of the PcFe-O₈ precursor (Supplementary Figure 26a). In our revised structural scheme (Figure 1a in the manuscript), we have assigned the

positions of the two iron centres.

3. We fully agree with the referee that the value of the magnetic moment in our work is low. However, the magnetization of $\text{K}_3\text{Fe}_2[\text{PcFe-O}_8]$ did not reach its saturated state even at the limited magnetic field strength by our previous facility (magnetization field: 7 T; temperature: 5 K, with the collaboration of Dr. Shengqiang Zhou group, Helmholtz-Zentrum Dresden-Rossendorf, Dresden, Germany). This was probably caused by the fractional ferromagnetic exchange interactions between nanoscale superparamagnetic clusters. Therefore, it is expected that the magnetic moment per Fe-site after saturation in $\text{K}_3\text{Fe}_2[\text{PcFe-O}_8]$ will be definitely higher than $0.71 \mu\text{B}/\text{Fe}$.

4. We also agree with the referee that there are two magnetic phases in $\text{K}_3\text{Fe}_2[\text{PcFe-O}_8]$. In order to further testify the ferromagnetic behavior and the magnetic mechanism, we re-checked different batches of synthetic $\text{K}_3\text{Fe}_2[\text{PcFe-O}_8]$ with the support of Prof. Bernd Büchner group (Leibniz Institute for Solid State and Materials Research Dresden, Germany) by SQUID measurements. Figure R11a shows ZFC and FC magnetization for one new batch of $\text{K}_3\text{Fe}_2[\text{PcFe-O}_8]$ in an applied magnetic field of 50 Oe. The M - T curves also demonstrate that $\text{K}_3\text{Fe}_2[\text{PcFe-O}_8]$ possesses superparamagnetic particles with blocking temperatures even above room temperature. Same as our previous M - T data (Figure 4a in the manuscript), there is a large difference between the ZFC and FC curves in the low temperature range. The ZFC curve (blue) shows a broad maximum about 50-60K, while the FC curve (red) exhibits a hysteresis due to the frozen spins under 50 Oe. There may be a competition between ferromagnetic and anti-ferromagnetic exchange interactions leading to spin-glass behavior. This reveals structural inhomogeneities in $\text{K}_3\text{Fe}_2[\text{PcFe-O}_8]$, which conveys that randomly oriented magnetic domains are more difficult to align in the MOF sample, thus leading to the co-existence of superparamagnetic and anti-ferromagnetic phases. In addition, the summit peak in the ZFC curve is relatively narrowed compared with that in our previous M - T data (Figure 4a in the manuscript), which is a potential evidence that the magnetic behavior

depends on the size of magnetic crystalline particles. Figures R11b and c show the M - H hysteresis curves at 1.8 K and 300 K. At 300 K, $\text{K}_3\text{Fe}_2[\text{PcFe-O}_8]$ retains a magnetic hysteresis with a saturation magnetization of $\sim 0.26 \mu_B/\text{Fe-site}$. Upon cooling by 1.8 K, an obvious increase in the hysteresis loop is evident that the ferromagnetic phase dominates at low temperature. However, the hysteresis still did not reach the saturation status with magnetization of $\sim 0.83 \mu_B/\text{Fe-site}$. Nevertheless, the contrast SQUID measurements on different batches of $\text{K}_3\text{Fe}_2[\text{PcFe-O}_8]$ further solidify its room-temperature ferromagnetism feature within magnetic clusters.

Figure R11 | Magnetic properties of the $\text{K}_3\text{Fe}_2[\text{PcFe-O}_8]$. **a**, Zero-field-cooled (ZFC) and field-cooled (FC) magnetization for $\text{K}_3\text{Fe}_2[\text{PcFe-O}_8]$ in an applied DC magnetic field of 50 Oe; **b**, Magnetic hysteresis loops obtained at 1.8 K and 300 K. **c**, Enlarged hysteresis loops in **b** image. **d**, Magnetic ground state in $\text{K}_3\text{Fe}_2[\text{PcFe-O}_8]$.

Therefore, based on the ^{56}Fe Mössbauer analysis (Figure R7) as well as the support from the

XPS (Supplementary Fig. 5c) and spin density distribution calculation (Fig. 2d in the manuscript), we can infer that the existence of two magnetic phases readily originates from different spin lengths localized at two kinds of Fe centres, as shown in Figure R11d. On the other hand, the polycrystalline feature by regarding to the varied crystalline domain sizes and the randomly oriented magnetic clusters gives rise to the heterogeneities. In addition, it is proposed that the magnetic coupling is induced by indirect exchange interaction between the localized iron spins via the highly delocalized π electrons along the fully conjugated backbones, thus enabling strong hybridization between the d/p orbitals of Fe, the Pc core, and the Fe-O₄ nodes (seen in Fig. 2c). As mentioned before, the calculated magnetic ground states of AA-serrated K₃Fe₂[PcFe-O₈] present an energy-favorable ferromagnetic coupling with a positive exchange coupling energy of 300 meV (Supplementary Fig. 17), which further validates this conclusion.

Even so, we also agree with the referee that possible existence of defects exist in polycrystalline K₃Fe₂[PcFe-O₈], which is generated from the grain boundaries, crystallite tilting as well as the edges of heterogeneity. This is the inevitable drawback of solution-based preparation of MOF materials. On the other hand, perfect single crystalline magnetic layered MOF materials (bulks and film) without defects are always the goal for MOF community, which could provide more intrinsic and fundamental understanding of the structure-magnetic properties relationship. Meanwhile, the control and engineering the defects is also a fantastic way to fabricate MOF materials with multi-functions.

ACTION:

We have added the above discussion in our revised manuscript and SI.

Comment 3. *Finally, well-characterized layered conductive magnets based on MOFs have been recently characterized (see Nature Chem. 10, 1056–1061 (2018)).*

Response: We have added this literature in our revised manuscript. We submitted our manuscript on 28 August 2018 while this report was published on 10 September 2018. We feel very grateful that this work from Pedersen, Long and Clérac et al. has benchmarked the research field towards conductive and magnetic MOFs.

Reviewer #3

General Comment: This manuscript describes the synthesis and characterization of a new 2D metal-organic framework (MOF) that displays ferromagnetic behavior at room temperature. The discovery of such room-temperature ferromagnetics could potential enable alternative computing architectures to solve the Moore's Law problem based on spintronics. Many ferromagnetic materials are known, most of which are inorganic and that require cryogenic temperatures. Coordination polymers and in particular MOFs have recently become of interest due to the high degree of design flexibility they possess. Consequently, this report showing ferromagnetic behavior with $T_c > 300$ K is an important discovery. The manuscript describes extensive characterization of the material, including spectroscopic, magnetic, and charge transport manuscript. It is also well written and for the most part reasonably understandable to a general audience. I recommend publication assuming the following points are addressed:

Response: We appreciate the Referee#3 for the encouraging comments and the positive recommendation for publication after revision. According to the reviewer's valuable suggestions, we have made additional effort to solidify our manuscript towards the structural, charge transport as well as the magnetic characterizations. Hope that our revision can address your concerns appropriately.

Comment 1. The material is not charge neutral and requires potassium cations for charge balancing. Presumably these are located within the MOF pores. Most likely because of this, the porosity of the material is not very high (75 m²/g, which is much lower than most MOFs). It would be interesting to know if the potassium ions could be exchanged for smaller Li⁺ or Na⁺ to increase the porosity.

Response: Thanks for the constructive suggestion from the referee. Here, we re-measured the N₂ sorption after activation of the K₃Fe₂[PcFe-O₈] powder. Before the BET measurement, the powder sample was treated in a supercritical CO₂ dryer for 5 days, and then activated at 80 °C overnight. As shown in Figure R12a (red), the newly resultant BET surface area of K₃Fe₂[PcFe-O₈] is about 206 m²/g. A pore distribution curve shows the average microporous pore size as ~ 1.4 nm, which is less than the calculated value (~ 1.8 nm), due to the incorporation of K⁺ ions inside the pores (Figure R12b, red line). The larger pore sizes of 25 nm and 35 nm, respectively, estimated from the pore distribution originate from the aggregates of the MOF nanoparticles. Following the suggestion from the referee, by changing the K⁺ ions with smaller Li⁺ ions (Li_xFe₂[PcFe-O₈]), the BET surface area for Li_xFe₂[PcFe-O₈] was enhanced to be 343 m²/g (Figure R12a, blue line) and

the average microporous pore size was determined as ~ 1.45 nm (Figure R12a, blue line), which further revealed the blocking influence of the incorporated counter ions on the porosity.

We have added this figure and the corresponding discussion in our revised manuscript and SI (Supplementary Fig. 7).

Figure R12 | Porous analysis of K₃Fe₂[PcFe-O₈] and Li_xFe₂[PcFe-O₈]. (a) N₂ adsorption isotherm measured at 77 K after supercritical CO₂ drying. (b) Pore size distribution profile calculated from the adsorption data using the DFT model.

Comment 2. Related to this point is the limited information concerning the activation procedure used. The methods section (p. 17) says that the “mixture” was “filtrated and soaked in degassed H₂O for 1 day.” By mixture, are the authors referring to a powder that formed and is suspended in the reaction solvent? Following the soaking, it is stated that the material was dried under vacuum. For how long was this done and at what temperature? If my presumption that this was done at room temperature is accurate, it seems likely that some water remains within the pores. The FTIR (Fig S2) suggests this is small, but Figure S5 shows an XPS peak assigned potentially to H₂O. Consequently, it would be good to rule out the effects of residual water by activating at a higher temperature (100 °C is preferable. If the material won’t tolerate that then at least something above 25 C) and then remeasuring the conductivity.

Response: We are grateful to the referee for pointing it out. We also feel sorry for the unclear description of the activation procedure and the inadequate presentation of the structural and compositional analysis for K₃Fe₂[PcFe-O₈] in our previous version. Here, we would like to address your questions and comments point by point.

1. The mixture is referred to the K₃Fe₂[PcFe-O₈] powders that formed and suspended in the reaction solvent. After filtration, the mixture was soaked in the degassed water for 1 day to remove

extra KOAc salts, followed by washing with acetone. Then the obtained $K_3Fe_2[PcFe-O_8]$ was dried under vacuum at 40 °C. From the FTIR results (Supplementary Figure 2), we can observe a small peak assigned to the water component. In the O_{1s} core level spectrum (Supplementary Fig. 5d), the peak at 533.5 eV can be assigned to the coordination H_2O (Chem. Soc. Rev. 2013, **42**, 5833-5857; Angew. Chem. Int. Edit. 2013, **52**, 8151-8155). Therefore, we fully agree with the referee that the FTIR and XPS results implied H_2O molecules incorporated in $K_3Fe_2[PcFe-O_8]$.

2. Following the suggestion from the reviewer, the TGA and the elemental analysis of $K_3Fe_2[PcFe-O_8]$ were performed to investigate the exact composition of $K_3Fe_2[PcFe-O_8]$. As shown in Figure R13, the TGA spectrum shows that the $K_3Fe_2[PcFe-O_8]$ starts to desolvate H_2O at 353 K (80 °C). A weight loss of 4.11% is an evidence of the coordinated water in $K_3Fe_2[PcFe-O_8]$. This result conveys the existence of ~2 coordination H_2O , leading to a chemical formula for this synthetic layered MOF as $K_3Fe_2[PcFe-O_8] \cdot 2H_2O$. In order to check the validity of our $K_3Fe_2[PcFe-O_8] \cdot 2H_2O$ formula, we have performed elemental analysis by a combination of inductively coupled plasma optical emission spectrometry (ICP-OES) and C, H, O and N combustion method; the results (in percentage) are presented in the table below:

	Fe(%)	C(%)	H(%)	N(%)	K(%)	O(%)
Found	17.39	40.16	1.38	11.83	12.13	17.05
Calculated	17.57	40.32	1.27	11.75	12.30	16.78

From these estimates, and taking into account of the formula $C_{32}H_{12}Fe_3K_3N_8O_{10}$, we can obtain the following elemental ratios: $C/Fe \approx 32/3$; $C/H \sim 32/13.2$; $C/N \approx 32/8$; $C/K \approx 32/3$; $C/O \approx 32/10.2$. Therefore, the element analysis validates a chemical formula as $K_3Fe_2[PcFe-O_8] \cdot 2.2H_2O$ ($C_{32}H_{12.4}Fe_3K_3N_8O_{10.2}$) for the layered 2D MOF that we have developed (named as $K_3Fe_2[PcFe-O_8]$ for short).

Figure R13 | TGA analysis of $K_3Fe_2[PcFe-O_8]$ under nitrogen atmosphere, indicating the existence of coordination H_2O in the layered MOF.

3. For the temperature-dependent conductivity measurement, the powder samples were heated at 100 °C under vacuum overnight and then the powder was compressed into pellets under the applied pressure of 1 GPa at 373 K (100 °C). Therefore, we can basically rule out the effect of water effect on the conductivity.

We have added the above discussion in our revised manuscript and SI (Supplementary Fig. 6).

Comment 3. *In general, nothing is said about the potential role of water and the counterions in the conductivity. The conductivity of materials such as Prussian Blue and its analogues is well known to depend on water in the pores.*

Response: Thanks for the constructive comment. We agree with the reviewer that the coordination solvents like water and the counterions will act as dopants to affect the conductivity of porous MOFs or other polymers (Polym. Int. 1999, **48**, 1080-1084; Chem. Mater. 2009, **21**, 1922-1926; Science 2014, **343**, 66-69; J. Am. Chem. Soc. 2017, **139**, 4175-4184; Nature Mater. 2018, **17**, 625-632). On the other hand, these dopants will definitely vary the crystallinity, the crystal size, the layer stacking model and the porosity, thus giving rise to the changes in the electronic structures, the carrier density and the charge mobility. It is worthy putting more effort and drawing another story for a systematic research, which will be also our future work. In our present work, we can basically rule out the effect of water on the conductivity of $K_3Fe_2[PcFe-O_8]$ due to the high-temperature (100 °C) drying before conductivity measurement.

Comment 4. Have the authors considered examining the effects of reduction or oxidation of the material as both a probe of the conductivity mechanism and as a potential strategy for tailoring the properties? Again referring to Prussian Blue, this is a mixed oxidation-state material with properties that must depend on the oxidation states of the two iron sites. Perhaps a simple experiment to test this idea could be added?

Response: Thanks for the valuable comments from this reviewer. We agree that the mixed oxidation states of two Fe ions contribute a lot to the conductivity of our MOF materials (Chem. Sci. 2017, **8**, 4450-4457; J. Am. Chem. Soc. 2018, **140**, 8526-8534; Nature Mater. 2018, **17**, 625-632). Previous report demonstrated that the conductivity of iron-contained MOF could be tuned by varying oxidation time by air exposure because the Fe^{2+} ions are extremely air sensitive (J. Am. Chem. Soc. 2018, **140**, 7411-7414). Therefore, following the suggestion from the reviewer, we remeasured the conductivity of $\text{K}_3\text{Fe}_2[\text{PcFe-O}_8]$ after oxidation by air for one week. As shown in Figure R14a, the conductivity after oxidation is 2-3 orders of magnitude lower than that before the oxidation at room temperature. At 350 K, the conductivity of $\text{K}_3\text{Fe}_2[\text{PcFe-O}_8]$ after oxidation is 3×10^{-5} S/m (2×10^{-3} S/m before oxidation). Figure R14b shows the Fe ($2p$) core level XPS spectrum for the oxidized MOF, which presents the coexistence of two Fe ions ($\text{Fe}^{2+}/\text{Fe}^{3+}$). The ratio of $\text{Fe}^{2+}/\text{Fe}^{3+}$ is calculated to be $\sim 1/3$ (the ratio was $\sim 2/1$ before oxidation, shown in the manuscript). From this contrast experiment, we can infer that the conductivity dropped rapidly after oxidation, due to the increasing content of Fe^{3+} .

We have added this contrast experiment in our revised manuscript and SI (Supplementary Fig. 22).

Figure R14 | Contrast experiments after air-oxidation. a) Variable-temperature electrical conductivity of $\text{K}_3\text{Fe}_2[\text{PcFe-O}_8]$ before (in red) and after oxidation (in blue) through van der Pauw method; b) Core level spectrum of Fe ($2p$) of $\text{K}_3\text{Fe}_2[\text{PcFe-O}_8]$ after oxidation.

Comment 5. *The reasons for using MOFs as a strategy for creating new ferromagnetics are primarily the exceptional synthetic flexibility. However, the authors don't say anything about the potential advantages of having nanoporosity as well as the desired magnetic properties. Since the term MOF is synonymous with a porous material and will no doubt attract attention because of this, the authors should make some comment regarding potential advantages and uses of the porosity.*

Response : Thanks for the insightful comment from this reviewer. Following the suggestion, we have added a brief description about the potential advantages and uses of the porosity:

“The porosity of layered MOFs enables the incorporation of guest molecules, which behave as a charge transfer manner to the metal nodes or organic linkers in MOFs. Therefore, the physical properties (like conductivity, magnetism, etc.) of MOFs materials could be further tuned by doping with guest molecules (Science 2014 **343**, 66-69; J. Am. Chem. Soc. 2016, **138**, 10088-10091; J. Am. Chem. Soc. 2018, **140**, 3871-3875; Chem. Sci. 2018, **9**, 4477-4482)”.

Comment 6. *Regarding prior reports of MOF ferromagnetics, I'm puzzled as to why the authors omit referencing their own 2018 paper in this journal describing a coronene-based MOF that is also ferromagnetic (R. Dong et al. NATURE COMMUNICATIONS | (2018)9:2637 | DOI: 10.1038/s41467-018-05141-4). Although the T_c of this material isn't reported for some reason, it must be at or below 20 K, so it doesn't compare with the material described in this manuscript. Nevertheless, it seems odd not to reference it in the intro and include it in Table S2.*

Response : Thanks for the reviewer's kind reminding. We feel sorry for omitting this work from our side. As suggested, we had already added it in our revised version (Supplementary Table 2).

Comment 7. *Figure 2 parts b and d: it is not easy to readily identify the spin isosurfaces because their color (white/gray) does not show up well against the white background. I suggest using a darker background or else different color scheme to make this easier. It would also be nice if the PDOS curves could be made larger so that it is easier to distinguish the colors of the various lines. Perhaps including larger versions in the SI would be an option.*

Response: Thanks for the valuable comment from this reviewer. To make the iso-surfaces more readable, we have changed its color into dark red in the revised manuscript (Figure 2b and 2d). Meanwhile, we have made a larger version of PDOS curve in revised Figure 2a and 2c as well as in updated SI (Supplementary Fig. 14).

Comment 8. In figure 2a and c, what are the dashed black lines supposed to indicate? It also a bit unclear what is meant on p. 10 line 176 by the statement “as shown in Fig. 2a”. this isn't obvious, so a little more explanation would be helpful.

Response : We feel very sorry for the unclear description in the former version. The dashed lines in Figure 2a and 2c indicate the spin-up bands and solid lines indicate the spin-down bands of $K_3Fe_2[PcFe-O_8]$. As shown in Figure 2a, the states near Fermi level are mainly derived from the delocalized π orbitals including the Fe(p), C(p), O(p) and N(p) orbitals. The density iso-surfaces indicated that the spin-density is mainly located on the Fe ions, while the spin polarization of bridging units (Fe-O₄) can be attributed to the polarization of these delocalized π orbitals from Fe(p), C(p), O(p) and N(p). We have added a brief description in the revised manuscript.

Comment 9. What is the explanation for the biexponential temperature dependence of the conductivity? The magnitude of the activation energy is attributed to grain boundaries, but why two different slopes?

Response : We thank the reviewer for pointing out this interesting question. Here, we ascribed the transport mechanism to thermally activated hopping, due to the heterogeneities in the MOF sample by taking into account of the polycrystallinity, the dispersed crystal sizes and the grain boundaries. The variable-temperature conductivity data for $K_3Fe_2[PcFe-O_8]$ were fit assuming an Arrhenius temperature dependence to determine the activation energy for charge hopping (Figure 3a, inset, in the manuscript). However, fitting the data with a single line revealed that the $\ln(\sigma)$ vs $1/T$ plot is not perfectly linear, indicating that the activation energy changes with temperature. It was possible to fit the high and low temperature data independently to extract activation energy values of ~ 261 and ~ 115 meV, respectively. As such, the Fermi level in the electronic structure will be also shifted with varying the temperature, leading to the changes in the band structures. Therefore, we extract two activation energy in high and low temperature regions for clarifying the temperature-dependent conductivity (Chem. Soc. Rev. 2012, **41**, 115-147; Nat. Commun. 2015, **6**, 7408-7416).

We have added this explanation in our revised manuscript.

Comment 10. Page 13: the mobility value looks pretty good but it would be good to compare it with other 2D materials. Another interesting aspect is that the bandgap of this material is only

0.63 eV, whereas the inorganic materials in table S1 are all 1 eV or greater. Would this be an advantage for device applications?

Response: Thanks for the keen comments from this reviewer. We have compared the carrier mobility with those of other layer stacked 2D coordination polymers and MOFs (as shown in the following Table R1). To be frank, the current comparison is very premature due to the utilization of various physical characterization methods for the estimation of mobility as well as the different sample forms. The precise comparison with well known inorganic 2D materials is not proper at this stage, which requires the synthesis of single crystalline 2D MOF or exfoliation of layered 2D MOFs into single-layer nanosheets. Regarding to the advantages of the narrowed band gap, it is generally up to the application targets. As a kind of electroactive layer for the DC electronics, narrowing the band gaps or increasing the orbital overlapping along 2D directions will be beneficial for higher charge mobility in MOF materials (*Angew. Chem. Int. Ed.* 2016, **55**, 3566-3579; *Nature Mater.* 2018 **17**, 1027-1032). However, it cannot cover every device function. For instance, the photodetector requires the desired band gaps for photo activation.

We have added the table in our revised SI (Supplementary Table 4).

Table R1 | Comparison of carrier mobility with those of other layer stacked 2D coordination polymers and MOFs

Name	Sample Form	Mobility (cm ² /Vs)	Method	Reference
K₃Fe₂[PcFe-O₈]	Powder	15±2	TRTS	This work
	Pellets	~0.1	Van der Pauw	
PiCBA	Thin films	0.005	TRTS	Angew. Chem. Int. Ed. 56 , 3920-3924, (2017).
Fe₃(THT)₂(NH₄)₃	Thin film	211±7	TRTS	Nat. Mater. 17 ,
		229±33	Hall effect	1027-1032, (2018)
Ni₃(HATP)₂	Thin Film	48.6	FET	J. Am. Chem. Soc. 139 , 1360-1363, (2017)
Cu-BHT	Thin Film	116 (electrons)	FET	Nat. Commun. 6 ,
		99 (holes)		7408-7416, (2015)
KxFe₂(BDP)₃	Single-crystalline	0.02(intrinsic)	FET	Nat. Mater. 17 , 625-632

	particle	0.29(reduction)		(2018)
Zn₂(TTFTB)	powder	0.2	FP-TRMC	J. Am. Chem. Soc. 134, 12932-12935 (2012).

THT = 2,3,6,7,10,11-hexathioltriphenylene; BHT = benzenehexathiol; HATP = 2,3,6,7,10,11-hexaaminotriphenylene. TRTS = time-resolved terahertz spectroscopy. FET = field effect transistor. FP-TRMC = flash photolysis-time-resolved microwave conductivity

Reviewers' comments:

Reviewer #1 (Remarks to the Author):

The authors performed additional measurements, which well addressed the comments. The presentation is also improved. I recommend it for publication.

Reviewer #2 (Remarks to the Author):

After the comments to the original manuscript, I still have important concerns about the magnetic nature of the system. Taking into account the magnetic properties and their interpretation of the data, it seems that there are several magnetic phases in their compound (see lines 305-3014, where the authors speculate about the possibility of antiferromagnetic interactions and/or spin-glass behaviour). Therefore, I consider that it is still too premature to publish the results in any journal.

In fact, the claim of ferromagnetism is not fully consistent with the experimental data. My main concern is that the saturation values at low temperature does not correspond to those expected for a combination of FeIII/FeII (with ferromagnetic or antiferromagnetic interactions in both High-Spin (HS) and Low-Spin (LS) scenarios). Following the structure of the Figure 1, every 3 Fe centers there are 2 FeII and 1 FeIII. Assuming the antiferromagnetic interactions shown in Fig. R.11.D, then the saturation values in Bohr magnetons per Fe center should be 1 (HS-HS), 5/3 (LS-HS), 7/3 (HS-LS) or 1/3 (LS-LS). However, they do not observe that values and, therefore, the authors must be measuring something different as the system presented in Fig 1. The possibility of a spin-glass scenario or superparamagnetic nanoparticles, as proposed in lines 305-314, should be further verified. All these possibilities are still open, but what is clear is that the material cannot be a ferromagnetic system as claimed in the title.

The authors can try to fit the magnetic data to different hamiltonians in order to test the possible spin scenario or, since they show TEM images with high-resolution, measure it with light polarization in order to observe from where the magnetic behavior arises.

Moreover, the spin texture proposed in Fig. R.11.D is a ferrimagnet and not a ferromagnet.

Although this is a minor comment, the system is not a genuine 2D material. Even if in the chemistry community the term "2D" is often taken as synonym of a layered compound ("layered = 2D"), this is not true in the physical one. And even in the chemistry community layered and 2D terms are currently used in different ways when dealing with graphene and other 2D materials. In fact, this distinction is made in recent papers dealing with 2D MOFs/coordination polymers. Therefore, for a broad audience as the one expected for a journal as Nature Communications the term 2D should be restricted to define a single layer of a material to avoid confusion.

Reviewer #3 (Remarks to the Author):

I reviewed the revised manuscript and SI and in general find that the points raised by the reviewers have been addressed satisfactorily. In fact, the authors have done a remarkably thorough job revising the manuscript, including new text, several new or revised figures, and extensive additional data. In particular:

- Questions regarding magnetic impurities were addressed using elemental analysis, HRETEM, XANES, and EXAFS, responding to the concern that a minor impurity could be responsible for the observed magnetic behavior. This cannot be completely ruled out if indeed a 1% impurity could produce the same effects, as suggested by Reviewer 2. However, there is essentially no evidence from the data reported that impurities are a factor.
- The activation procedures were substantially improved by employing supercritical CO₂ to more

thoroughly remove impurities. The results indicate higher surface areas, as expected. In addition, experiments were done using the Li salt (instead of K), which show a higher surface area, consistent with these ions occupying the pores and reducing the surface area.

- An issue regarding the use of “2D material” terminology was addressed adequately, in my opinion. The authors are correct that the MOF community uses this wording to refer to layered materials with minimal interlayer interaction, so the modifications to the text in this regard are appropriate.

A few minor comments regarding the response to the three reviewers:

Reviewer 1:

- Comment 5: I’m not sure what is meant by “sequentially” in referring to the sample cooling process.
- Comments 6 and 7: the figure numbers here are incorrect, so the authors should check to make sure they are numbered correctly in the text and SI.

Reviewer 2:

- Comment 1: I agree with the authors that it is very difficult to rule out the possibility of amorphous phases. The PXRD data (SI Fig. 8) do not look excessively broad, which might indicate a high percentage of noncrystalline material. It should also be noted that controlling the stacking arrangement of these layered materials is difficult, which can add some ambiguity to the interpretation.
- Also concerning comment 1: the detailed text in the author’s response to this comment is helpful in understanding how the authors reached their conclusion. I suggest that some of this be included in the SI to assist readers if they have the same question.
- Comment 2 (p. 18 of the response): the authors might want to make a comment regarding the possible existence of defects; I didn’t see anything in the manuscript about this.

Reviewer 3:

- Many of the comments by this reviewer were addressed in the response to the other reviewers.
- The experiment of aging the material in air, which showed that the conductivity decreased after 1 week, addresses the question about probing the mechanism and tailoring the properties. The fact that the conductivity decreased suggests that the resulting, more highly oxidized, components are insulating. Assuming that the magnetic and charge-conducting properties are due to the same material, this result would seem to support the conclusion that iron oxide impurities do not play a major role (Fe₂O₃ isn’t magnetic or conducting).

Detailed response to the comments from the reviewers

Reviewer #1 (Remarks to the Author):

General comment: The authors performed additional measurements, which well addressed the comments. The presentation is also improved. I recommend it for publication.

Response: We appreciate the Reviewer#1 for the positive comments and recommendation for publication.

Comment 5: I'm not sure what is meant by "sequentially" in referring to the sample cooling process.

Response: We are sorry for the unclear description in referring the sample cooling process. "Sequentially" means that the MOF sample was cooled down continuously, and the rate of this process was 35 K/min. We have clarified the description in the revised manuscript.

Comments 6 and 7: the figure numbers here are incorrect, so the authors should check to make sure they are numbered correctly in the text and SI.

Response: Thanks for the reviewer's kind reminding. We have double checked the figure numbers and corrected the mistake in the revised manuscript.

Reviewer #2 (Remarks to the Author):

General comment: *After the comments to the original manuscript, I still have important concerns about the magnetic nature of the system. Taking into account the magnetic properties and their interpretation of the data, it seems that there are several magnetic phases in their compound (see lines 305-3014, where the authors speculate about the possibility of antiferromagnetic interactions and/or spin-glass behaviour). Therefore, I consider that it is still too premature to publish the results in any journal. In fact, the claim of ferromagnetism is not fully consistent with the experimental data. My main concern is that the saturation values at low temperature does not correspond to those expected for a combination of FeIII/FeII (with ferromagnetic or antiferromagnetic interactions in both High-Spin (HS) and Low-Spin (LS) scenarios). Following the structure of the Figure 1, every 3 Fe centers there are 2 FeII and 1 FeIII. Assuming the antiferromagnetic interactions shown in Fig. R.11.D, then the saturation values in Bohr magnetons per Fe center should be 1 (HS-HS), 5/3 (LS-HS), 7/3 (HS-LS) or 1/3 (LS-LS). However, they do not observe that values and, therefore, the authors must be measuring something different as the system presented in Fig 1. The possibility of a spin-glass scenario or superparamagnetic nanoparticles, as proposed in lines 305-314, should be further verified. All these possibilities are still open, but what is clear is that the material cannot be a ferromagnetic system as claimed in the title. The authors can try to fit the magnetic data to different hamiltonians in order to test the possible spin scenario or, since they show TEM images with high-resolution, measure it with light polarization in order to observe from where the magnetic behavior arises. Moreover, the spin texture proposed in Fig. R.11.D is a ferrimagnet and not a ferromagnet.*

Response: We appreciate the referee#2 for the valuable comments and the critical insights on the magnetic analysis. We feel also sorry for our overly claimed in the previous submission and agree with the suggestion from the referee#2. As we exclude the major influence of magnetic impurities, such as FeO and Fe₂O₃, by our data based on the combined state-of-art analysis by XPS, XANES, EXAFS, ⁵⁶Fe Mössbauer spectra, TGA, elemental analysis and HR-TEM, we have revised our conclusion about the magnetic data according to the suggestion: *Magnetization experiments and ⁵⁷Fe Mössbauer spectroscopy demonstrate the presence of magnetic ordering in K₃Fe₂[PcFe-O₈] arising from the magnetic coupling between iron centers via delocalized π electrons. The sample exhibits superparamagnetic features due to a distribution of the crystal size and possesses magnetic hysteresis up to 350 K.*

Thus, in this work, we establish a new semiconducting layer-stacked MOF with Lieb lattice and in-plane full π-d conjugation, and achieve a high charge mobility and a superparamagnetic behavior. We hope our additional efforts appropriately address your concerns.

The detailed revision in our manuscript was listed as below, which was highlighted in red color:

1. Title: we have appropriately revised the title as “A Semiconducting Layered Metal-Organic

Framework Magnet”, which avoids the overly stated “2D” and “room temperature ferromagnetism”.

2. Abstract: the conclusion about “*Magnetization experiments and ^{57}Fe Mössbauer spectroscopy demonstrate the presence of magnetic ordering in $\text{K}_3\text{Fe}_2[\text{PcFe-O}_8]$ arising from the magnetic coupling between iron centers via delocalized π electrons. The sample exhibits superparamagnetic features due to a distribution of crystal size and possesses magnetic hysteresis up to 350 K.*” has been presented.

3. Page 5: the conclusion sentence about “*Moreover, this $\text{K}_3\text{Fe}_2[\text{PcFe-O}_8]$ presents magnetic ordering in nanoscale crystallites, which overall shows superparamagnetism with a broad distribution of blocking temperatures.*” has been added.

4. Page 5: the perspective sentence about “*By further optimizing the crystalline quality and increasing the crystallite size, our work presents the possibility to achieve room-temperature ferromagnetism in a semiconducting layered MOF, highlighting the potential for developing a new generation of MOFs-based spintronics.*” has been added.

5. Figure 4 has been revised: the ^{57}Fe Mössbauer spectrum of $\text{K}_3\text{Fe}_2[\text{PcFe-O}_8]$ measured at 25 K was added as Figure 4d. Thus, new Figure 4 is shown as the following.

Figure 4 | Magnetic properties of the $\text{K}_3\text{Fe}_2[\text{PcFe-O}_8]$. a, Magnetic hysteresis loops obtained at

different temperatures for $K_3Fe_2[PcFe-O_8]$; **b**, Zero-field-cooled (ZFC) and field-cooled (FC) magnetization for $K_3Fe_2[PcFe-O_8]$ in an applied DC magnetic field of 100 Oe; **c** and **d**, ^{57}Fe Mössbauer spectra of $K_3Fe_2[PcFe-O_8]$ measured at a temperature of 300 K and 25 K, respectively.

6. Pages 15-16: a description for magnetization measurement at 5 K in Figure 4a has been revised as *“Upon cooling to 5 K, an obvious increase in the coercivity and the magnetization was observed. Particularly, the hysteresis still did not reach the saturation at 7 T with magnetization of $\sim 0.7 \mu_B/Fe$ -site at 5 K. These observations suggest the superparamagnetic nature in the sample.”*

7. Page 16: a description about Figure 4b has been revised as *“The blocked superparamagnetic nature is also confirmed by the zero field cooled (ZFC) and field-cooled (FC) measurements (Fig. 4b). The FC magnetization in an external field of 100 Oe reveals a monotonic decrease upon increasing temperature. The corresponding ZFC magnetization at the same external field increases with temperature with a broad summit peak around 200 K, approaching the FC curve at around 300 K. The irreversibility between the ZFC and FC curves is due to blocked superparamagnetic clusters which have a broad distribution of blocking temperatures. This is the result of structural inhomogeneities in $K_3Fe_2[PcFe-O_8]$ (shown in the TEM images, Supplementary Fig. 10 and 11).”*

8. Page 17: a sentence about *“However, the polycrystalline nature of the samples (10-100 nm in domain size) renders the observed superparamagnetism, which could be addressed by improving the sample preparation, such as to obtain single crystals in the future.”* has been added.

9. Page 18: a description about the Mössbauer measurements (Figures 4c and d) has been added as *“The Mössbauer measurements suggest the superparamagnetic nature of the sample. In the superparamagnetic state, the magnetization direction of nanoparticle fluctuates among the easy axes of magnetization when there is no external magnetic field. The relaxation time depends on the size of the particles and the temperature. Therefore, magnetically, three sites are observed at 25 K because some particles are large enough to have a longer enough relaxation time, which contributes to the sextet, while the remained randomly oriented particles are superparamagnetic (red site A).”*

10. The scheme illustrating the magnetic mechanism in Supplementary Figure 27 has been

deleted.

11. Two important literatures have been cited: ref15. López-Cabrelles, J. *et al.* Isoreticular two-dimensional magnetic coordination polymers prepared through pre-synthetic ligand functionalization. *Nat. Chem.* **10**, 1001-1007 (2018). ref19. Pedersen, K. S. *et al.* Formation of the layered conductive magnet $\text{CrCl}_2(\text{pyrazine})_2$ through redox-active coordination chemistry. *Nat. Chem.* **10**, 1056-1061 (2018).

Comment 1: *Although this is a minor comment, the system is not a genuine 2D material. Even if in the chemistry community the term “2D” is often taken as synonym of a layered compound (“layered = 2D”), this is not true in the physical one. And even in the chemistry community layered and 2D terms are currently used in different ways when dealing with graphene and other 2D materials. In fact, this distinction is made in recent papers dealing with 2D MOFs/coordination polymers. Therefore, for a broad audience as the one expected for a journal as Nature Communications the term 2D should be restricted to define a single layer of a material to avoid confusion.*

Response: We fully agree with the critical suggestion on the definition of “2D” as a single layer from Referee#2. To make the nature of our MOF sample clear, we have completely removed the “2D” description but stressed that we achieved layered MOFs. The revision has been highlighted in red color in our main text.

Comment 2: *I agree with the authors that it is very difficult to rule out the possibility of amorphous phases. The PXRD data (SI Fig. 8) do not look excessively broad, which might indicate a high percentage of noncrystalline material. It should also be noted that controlling the stacking arrangement of these layered materials is difficult, which can add some ambiguity to the interpretation. Also concerning comment 1: the detailed text in the author’s response to this comment is helpful in understanding how the authors reached their conclusion. I suggest that some of this be included in the SI to assist readers if they have the same question.*

Response: We appreciate Referee#2 for pointing out the two critical questions: How to improve the crystallinity of the layered MOFs? How to control the stacking modes in layered MOFs? In this work, we define the structural nature of the resultant MOF by PXRD and HRTEMs: poly-crystalline powders with a distribution of domain sizes, square lattices with $a = b = 18.1 \text{ \AA}$ and AA-serrated layered structure. According to the comment from the referee, we have added a description about the questions on the crystallinity and stacking arrangement in the SI (Page 18), which is expected to help the readers achieve better impression on the controlled synthesis. The text is shown as following: The crystallinity and the structural determination (such as repeated

units, stacking pattern, amorphous structures and impurities) are the key research motivation towards the controlled synthesis of MOFs. As one of the common targets, it remains challenging to synthesize highly crystalline or even single-crystalline layer-stacked MOFs, which can be essential to achieve reliable structure-property relationship based on the pristine sample. Notably, a few successful examples about the magnetic layered MOFs have been recently reported by Coronado et al. (Nat. Chem. 2018, **10**, 1001-1007), Clérac et al. (Nat. Chem. 2018, **10**, 1056-1061), Harris et al. (J. Am. Chem. Soc. 2015, **137**, 15699-15702; J. Am. Chem. Soc. 2017, **139**, 4175-4184) and Long et al. (J. Am. Chem. Soc. 2015, **137**, 15703-15711; J. Am. Chem. Soc. 2018, **140**, 3040-3051). Here in this work, we synthesized a magnetic, polycrystalline, layer-stacked MOF, which comprises of poly-dispersed nanoscale crystallites. It is still difficult to exclude the possible presence of the amorphous phases from the current XRD and local HR-TEM characterizations. Nevertheless, we put our efforts to exclude the possible iron impurities in the resultant $K_3Fe_2[PcFe-O_8]$ by the above spectroscopy studies (Fig. 1, Supplementary Fig. 5). We also acquired a number of TEM images within different regions of the sample, as shown in Supplementary Fig. 10-11. Based on these TEM images, the rectangle-shaped nanocrystals of $K_3Fe_2[PcFe-O_8]$ are well visualized. In fact, our work indeed offers a high-quality structural resolution by TEM, which clearly presents nanoscale crystalline domains in $K_3Fe_2[PcFe-O_8]$ comprising square lattices.

Comment 3: (p. 18 of the response): the authors might want to make a comment regarding the possible existence of defects; I didn't see anything in the manuscript about this.

Response: Thank the referee for the comment. The grain boundaries generated from the nanoscale crystalline domains shall be a kind of defects, which generally block the extended conjugation and the long-range charge transport. We have presented the polycrystalline feature of the MOF sample in our manuscript by regarding to the varied domain sizes.

Comment 4: The revision plan may be acceptable. In any case, the authors should also be careful in the text and should introduce some sentences to point out the difficulty in characterizing the material as the results are still indicating that the compound is not a very clean (single) magnetic phase and surely a more deep characterization should be needed in the future to unambiguously exclude the possibility of magnetic impurities coming from iron.

Response: We really appreciate the Referee for agreeing with our revision plan. According to the

comment, we also pointed out the challenge and future research perspective about this research topic in the revised manuscript. A note has been added on Page 19: *“Nevertheless, it should be noted that further experiments are required to improve the crystalline quality to obtain ferromagnetic samples at or even above room temperature and sophisticated characterizations should be done to unambiguously exclude the possibility of magnetic secondary phases. Our work is expected to encourage more physical researches on magnetic and semiconducting properties of layered conjugated MOFs.”*

Reviewer #3 (Remarks to the Author):

General comment: *I reviewed the revised manuscript and SI and in general find that the points raised by the reviewers have been addressed satisfactorily. In fact, the authors have done a remarkably thorough job revising the manuscript, including new text, several new or revised figures, and extensive additional data. In particular:*

- *Questions regarding magnetic impurities were addressed using elemental analysis, HRETEM, XANES, and EXAFS, responding to the concern that a minor impurity could be responsible for the observed magnetic behavior. This cannot be completely ruled out if indeed a 1% impurity could produce the same effects, as suggested by Reviewer 2. However, there is essentially no evidence from the data reported that impurities are a factor.*
- *The activation procedures were substantially improved by employing supercritical CO₂ to more thoroughly remove impurities. The results indicate higher surface areas, as expected. In addition, experiments were done using the Li salt (instead of K), which show a higher surface area, consistent with these ions occupying the pores and reducing the surface area.*
- *An issue regarding the use of “2D material” terminology was addressed adequately, in my opinion. The authors are correct that the MOF community uses this wording to refer to layered materials with minimal interlayer interaction, so the modifications to the text in this regard are appropriate.*
- *Many of the comments by this reviewer were addressed in the response to the other reviewers.*
- *The experiment of aging the material in air, which showed that the conductivity decreased after 1 week, addresses the question about probing the mechanism and tailoring the properties. The fact that the conductivity decreased suggests that the resulting, more highly oxidized, components are insulating. Assuming that the magnetic and charge-conducting properties are due to the same material, this result would seem to support the conclusion that iron oxide impurities do not play a major role (Fe₂O₃ isn't magnetic or conducting).*

Response: We fully appreciate the Reviewer#3 for the encouraging comments and the positive recommendation for publication.

REVIEWERS' COMMENTS:

Reviewer #2 (Remarks to the Author):

The authors have satisfactorily addressed most of my concerns and have decreased their initial claims about ferromagnetism and now the behavior is associated to superparamagnetism. If that is so, there are still some contradictions that should be removed in the abstract and along the text. The main one is the assessment about magnetic ordering. This long range magnetic ordering has not been proved (specific heat measurements are needed to prove it); therefore, I should prefer to decrease this claim changing in the abstract (and along the text) "magnetic ordering" with "spontaneous magnetization" (in the first sentence) or "long-range magnetic correlations" (in the sentence "Magnetization experiments and Mössbauer spectroscopy demonstrate the presence of magnetic ordering").

With these small changes, I should consider the paper acceptable for publication.

Detailed response to the comments from the reviewers

Reviewer #2

General Comment: *The authors have satisfactorily addressed most of my concerns and have decreased their initial claims about ferromagnetism and now the behavior is associated to superparamagnetism. If that is so, there are still some contradictions that should be removed in the abstract and along the text. The main one is the assessment about magnetic ordering. This long range magnetic ordering has not been proved (specific heat measurements are needed to prove it); therefore, I should prefer to decrease this claim changing in the abstract (and along the text) "magnetic ordering" with "spontaneous magnetization" (in the first sentence) or "long-range magnetic correlations" (in the sentence "Magnetization experiments and Mössbauer spectroscopy demonstrate the presence of magnetic ordering").*

With these small changes, I should consider the paper acceptable for publication.

Response: We appreciate the Reviewer #2 for the constructive suggestions again and the recommendation for publication after minor revision. In the revised manuscript, we have replaced the “magnetic ordering” by "spontaneous magnetization" or "long-range magnetic correlations" in the abstract and along the text.